# Antifreeze protein dispersion in eelpouts and related fishes reveals migration and climate alteration within the last 20 Ma

Rod S. Hobbs[1], Jennifer R. Hall[2], Laurie A. Graham[3]*, Peter L. Davies[3], Garth L. Fletcher[1]

**1** Department of Ocean Sciences, Memorial University of Newfoundland, St John's, Newfoundland, Canada, **2** Aquatic Research Cluster, CREAIT Network, Memorial University of Newfoundland, St. John's, Newfoundland, Canada, **3** Department of Biomedical and Molecular Sciences, Queen's University, Kingston, Ontario, Canada

* grahamla@queensu.ca

**Data Availability Statement:** The DNA sequences are available from NCBI GenBank under accession numbers KR872957-KR872964. The

## Abstract

Antifreeze proteins inhibit ice growth and are crucial for the survival of supercooled fish living in icy seawater. Of the four antifreeze protein types found in fishes, the globular type III from eelpouts is the one restricted to a single infraorder (Zoarcales), which is the only clade know to have antifreeze protein-producing species at both poles. Our analysis of over 60 unique antifreeze protein gene sequences from several Zoarcales species indicates this gene family arose around 18 Ma ago, in the Northern Hemisphere, supporting recent data suggesting that the Arctic Seas were ice-laden earlier than originally thought. The Antarctic was subject to widespread glaciation over 30 Ma and the Notothenioid fishes that produce an unrelated antifreeze glycoprotein extensively exploited the adjoining seas. We show that species from one Zoarcales family only encroached on this niche in the last few Ma, entering an environment already dominated by ice-resistant fishes, long after the onset of glaciation. As eelpouts are one of the dominant benthic fish groups of the deep ocean, they likely migrated from the north to Antarctica via the cold depths, losing all but the fully active isoform gene along the way. In contrast, northern species have retained both the fully active (QAE) and partially active (SP) isoforms for at least 15 Ma, which suggests that the combination of isoforms is functionally advantageous.

## Introduction

Most marine teleosts are unable to inhabit ice-laden sea waters characteristic of polar and sub-polar oceans because the temperature of the water (−1.9°C) can be a full degree lower than the freezing point of their body fluids (−0.7 to −0.9°C) [1]. In contrast such environmental conditions pose no risk to most invertebrates as their freezing points are usually the same as that of sea water [2]. Teleost fish that can survive and thrive in such environments do so by producing antifreeze proteins (AFPs) or glycoproteins (AFGPs) that bind to nascent ice crystals within their body fluids, thereby preventing their further growth that would ultimately result in death

corresponding protein accession numbers are ALL26673-ALL26680.

**Funding:** This research was supported by the Canadian Institutes of Health Research (https://cihr-irsc.gc.ca/e/193.html)(Foundation Grant FRN 148422 to P.L.D.) and the Natural Sciences and Engineering Research Council (https://www.nserc-crsng.gc.ca/index_eng.asp) (Discovery Grant 6836-06 to G.L.F.). P.L.D. holds the Canada Research Chair in Protein Engineering (https://www.chairs-chaires.gc.ca/home-accueil-eng.aspx). The funders had no role in study design, data collection and analysis, decision to publish, or preparation of the manuscript.

**Competing interests:** The authors have declared that no competing interests exist.

[3–7]. The difference between the melting point and non-equilibrium freezing point is defined as thermal hysteresis (TH) and is a measure of AFP activity [5].

To date, three non-homologous, physiologically functional groups of AFPs (types I-III), as well as the AFGPs, have been described in a variety of fish taxa [8–10]. Both type II and type III AFP, are globular and non-repetitive, with the type II AFP gene having been horizontally transferred between three fish orders (Osmeriformes, Perciformes, and Clupeiformes) that diverged over 200 Ma ago [11, 12]. In contrast, both AFGP and type I AFP are repetitive proteins. AFGPs consist of variable numbers of glycosylated tripeptide repeats and arose by convergent evolution in Antarctic Notothenioids (Order Perciformes) and the unrelated northern cods (Order Gadiformes) from two different progenitor sequences [13–16]. The repetitive alanine-rich α-helical type I AFPs also arose by convergence, within four families from three orders: Pleuronectiformes, Labriformes and Perciformes [17]. All of the AFP-producing fish orders mentioned above diverged prior to the Eocene, which began with polar oceans that were ice-free during the Paleocene–Eocene thermal maximum at 55 Ma [18, 19]. Therefore, the impetus behind the evolution of these unique ice binding structures was the subsequent occurrence of sea ice following the onset of global cooling and glaciation [20]. These are clear examples of convergent protein evolution to a common function [21]. The occurrence of lateral gene transfer [12], and of convergence to highly similar repeat sequences [13, 17] adds a level of evolutionary complexity that is absent from most gene families.

In contrast to the other fish AFPs and AFGPs, type III AFP is restricted to a single taxonomic group, order Perciformes, infraorder Zoarcales (previously suborder Zoarcoidei), that diverged approximately 50 Ma ago [19] (Fig 1). This ~7 kDa protein arose from the C-terminal domain of sialic acid synthase (SAS) following a gene duplication [22–24]. The first type III AFPs were isolated from ocean pout (*Zoarces americanus*, family Zoarcidae) [25, 26]. They are found as mixtures of SP-Sephadex- (SP) and QAE-Sephadex-(QAE) binding isoforms that are only about 55% identical. Both of these isoform sets have the capacity to bind to ice, but only a subset of QAE isoforms are able to completely halt ice crystal growth [27, 28]. Interestingly, the SP isoforms can be made fully active in stopping ice growth if as little as 1% of the QAE isoform is included with them [28].

Fishes from the infraorder Zoarcales have a global distribution and are the only group of fishes that have been demonstrated to have AFP-producing species that have conquered the icy waters at both poles (Fig 1). Five families, four of which are restricted to Northern waters, were shown by early Southern blotting studies to possess numerous type III AFP genes: Zoarcidae [ocean pout (*Z. americanus*)]; Anarhichadidae [spotted wolffish (*Anarhichas minor*) and Atlantic wolffish (*A. lupus*)]; Pholidae [rock gunnel (*Pholis gunnellus*)]; Stichaeidae [radiated shanny (*Ulvaria subbifurcata*)] [29–31]; Cryptacanthodidae [wrymouth (*Cryptacanthodes maculatus*)] [32, unpublished data]. However, *AFP* sequences have only been obtained from sequences within Zoarcidae and Anarhichadidae [25–27, 29, 30, 33, 34]. Most of these species inhabit the shallow inshore waters of Newfoundland, where they are frequently exposed to subzero temperatures and ice during the winter months [35].

Type III AFPs have also been characterized from two species that reside in the frigid Southern Ocean around Antarctica. Both *Lycodichthys dearborni* and *Pachycara brachycephalum* are commonly called Antarctic eelpouts, so to differentiate them, we will refer to *L. dearborni* as Antarctic eelpout and *P. brachycephalum* by scientific name only. The AFP complement of the Antarctic eelpout has been studied through a combination of protein, cDNA and genomic DNA sequencing (yielding over 20 gene sequences). This species produces both monomers and tandemers consisting of two or more linked AFP domains [23, 36–40], whereas the

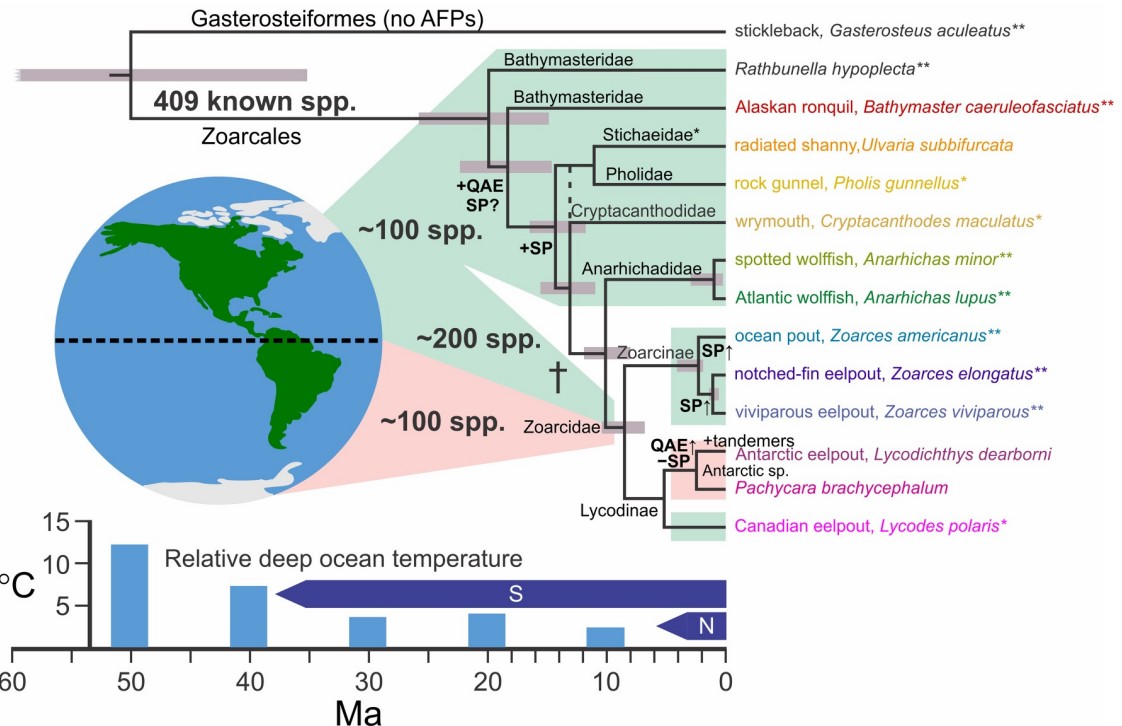

**Fig 1. Detailed relationships among species of the infraorder Zoarcales discussed in this study.** Divergence times were obtained from the following studies; Gasterosteiformes and Zoarcales, from 21 loci from a broad range of teleost fishes [19], Bathymasteridae, from 10 nuclear genes [88] and within Zoarcales [42] or Anarhichadidae [89] using fewer loci. A double asterisk denotes that the fish from these studies was the same as the species in this study, whereas a single asterisk denotes that the fish was a different species within the same genus. The 95% posterior density intervals for the branch points are shown by grey bars. Species names are coloured in a rainbow pattern from red to purple from earliest to latest time of divergence of each family. The location of the Cryptacanthodidae varies between studies (denoted with a dotted line) [19, 46] and some Stichaeidae cluster with Pholidae [44–46]. Information on Lycodinae and radiated shanny was limited. Therefore, their relationship to the others was determined using the *COX* gene with Alaskan ronquil as the outgroup (S2 Fig), with divergence times estimated from the nearest calibrated node. The nodes at which there is the earliest evidence for the presence (+) or absence (-) of SP or QAE isoforms are indicated, as well as branches along which gene duplications have occurred (SP↑). The number of known Zoarcales species and their range (green background for northern, pink for southern) was determined from FishBase [72]. The average temperature of the deep ocean, relative to the present day, as well as the temporal extent of northern (N) and southern (S) glaciation, was adapted from Fig 1 of [90].

northern species studied produce only monomers. Interestingly, there is no evidence of tandemers in the second Antarctic species examined, *P. brachycephalum* [39, 41].

Given the diverse evolutionary history of fish AFPs, we examined in this study several questions about type III AFPs. The first was to establish if this AFP has spread through the Zoarcales by direct descent as opposed to parallel evolution. By mining unique sequences from various databases and transcriptome studies, and through targeted sequencing projects on several species, we can confidently say that all known type III AFPs are related by descent. Secondly, we wanted to date the origin of the type III AFP, so we successfully expanded the search for *AFP* sequences in other Zoarcales families out to the Alaskan ronquil, *Bathymaster caeruleofasciatus*, which diverged around 18 Ma [42] (Fig 1). Finally, having established here that all type III AFPs are related by descent, we addressed the timing of the colonization of Antarctic waters by zoarcids. The most plausible explanation is one of recent migration of a founder species from the north that transitioned through the cold ocean depths linking the two poles.

## Results

### Insights into species relationships in the infraorder Zoarcales

A number of phylogenetic and taxonomic studies have elucidated the relationships amongst the infraorder Zoarcales and this information is summarized in Fig 1 [19, 43–46]. To confirm and bolster the connections within the tree of Zoarcales, (particularly for the polyphyletic Stichaeidae, the radiated shanny, and Antarctic eel pouts), *cytochrome oxidase (COX)* gene sequences from these (or closely related) species were downloaded and an alignment was generated (S1 Fig). This was used to produce a phylogenetic tree (S2 Fig). All of the branch points in the DNA-based tree were consistent with those determined in the taxonomic studies (Fig 1). Key points from this new study are that: i) the Antarctic species were found to be more closely related to each other than to the Canadian eelpout; ii) the radiated shanny, like some other Stichaeidae, was found to be more closely related to the rock gunnel than to any of the other species; iii) the three species newly-examined in this study are found in two lineages that diverged ~18 Ma (Alaskan ronquil) and ~15 Ma (radiated shanny and rock gunnel). The two families from which AFP sequences were previously known, Anarhichadidae and Zoarcidae, only diverged from each other ~10 Ma.

### Type III AFPs arose early in the infraorder Zoarcales

To trace the evolutionary history of type III AFP and date its origin, we set out to expand the number of known sequences from the three Zoarcales species mentioned above (Alaskan ronquil, radiated shanny and rock gunnel that are found in two lineages that diverged early in this clade. Full-length cDNA sequences encoding both QAE and SP isoforms were obtained from rock gunnel and radiated shanny using primers based on known sequence (S1 Table) and a combination of RLM-RACE and RT-PCR. When the protein sequences of these two species are aligned (Fig 2, S3 Fig), the SP isoforms are 72% identical while the QAE isoforms are 79% identical. This identity drops to ~60% when SP and QAE isoforms are compared. The accession numbers and characteristics of these sequences are given in S2 Table. QAE and SP isoforms were originally categorized by their ability to bind to positively-charged quaternary aminoethyl (QAE) or negatively-charged sulfopropyl (SP) resins at neutral pH [26]. However, the rock gunnel-Q1 sequence has a predicted isoelectric point at pH 9.5, more like that of SP isoforms. Therefore, sequence similarity is a better way to categorize type III AFPs. When the cDNA sequences were aligned with those of other fish (S4 Fig) and used to generate a phylogenetic tree (Fig 3), the SP sequences clustered together in this phylogeny, as did the *QAE* sequences, consistent with the relatedness of these two species (Fig 1). However, the first exon, which encodes the signal peptide, was identical between the rock gunnel SP and QAE isoforms. Phylogenetic analysis of this short exon (S5 Fig), using the homologous region of SAS, which does not encode a signal peptide [23], to root the tree, shows that for all other sequences, QAE sequences cluster, as do SP sequences. In contrast, the rock gunnel and radiated shanny exons cluster outside of these two groups. This suggests that exon shuffling may have occurred in the common ancestor of these two species.

Northern blotting was performed using various tissues from both the rock gunnel and the radiated shanny (S6 Fig) and the transcripts were most abundant in the liver. Moderate expression was found in skin, gill and stomach, indicating that that the liver-dominant expression originally observed in Zoarcidae [47] and Anarhichadidae [29] is also a feature that arose early in Zoarcales. A partial *type III AFP* sequence from an even more divergent species, the Alaskan ronquil (Fig 1), was amplified from genomic DNA in two overlapping fragments using semi-nested PCR (see Supplementary Methods). This single sequence encodes a QAE isoform,

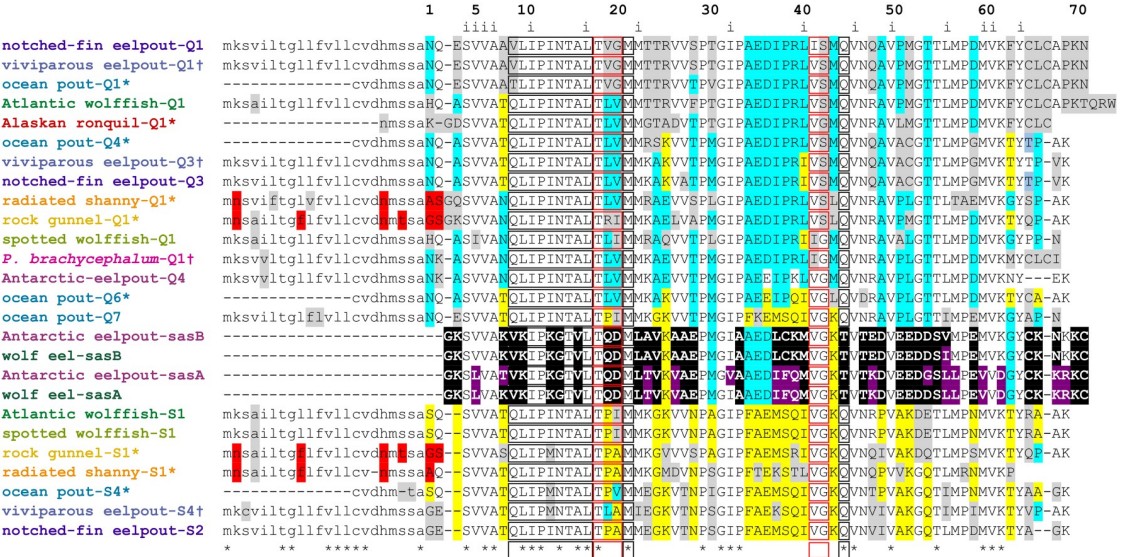

**Fig 2. Alignment of a representative selection of known type III AFP sequences.** Sequences are named using the common name for each species except for *P. brachycephalum*. Names are colored as in Fig 1 with an asterisk indicating the sequences were obtained by PCR in this study (S3 Table) and with a dagger for those assembled from the SRA database (S4 Table). They are numbered consecutively by species within the QAE (Q) or SP (S) group as they appear in S3 Fig. Residues conserved in all isoforms (sasA and B) are indicated with asterisks, variations characteristic of the QAE and SP groups are highlighted cyan and yellow respectively, with other variations in grey. Shared differences in the signal peptides of radiated shanny and rock gunnel are highlighted red. Black and red boxes show residues on the pyramidal- and prism-plane binding surfaces respectively [79]. Inward-facing residues (i) are indicated along the top. Residues unique to SAS are highlighted black with other differences within this group highlighted purple. Signal peptides are in lower case font. Dashes indicate gaps (internal) or incomplete sequence (at termini). The complete protein and nucleotide alignments are shown in S3 and S4 Figs, respectively.

Alaskan ronquil-Q1 (Fig 2). The gene structure is consistent with other *type III* genes. The 174 bp intron (not shown) lies close to the expected location, but both splice junctions are shifted leftward by three bp (S4 Fig). This sequence is most similar to Atlantic wolffish-Q2. The identities between the protein sequences, as well as between the coding sequences and the single intron, were between 94 and 95%. The 3' UTR is up to 97% identical to other *type III* sequences. SP sequences were not recovered, but their existence cannot be ruled out as the DNA amount and quality from the museum specimen was low and/or the primers used may not match Alaskan ronquil sequences. The presence of this sequence in the Alaskan ronquil pushes back the origin of type III AFP to approximately 18 Ma.

## Additional AFP variants in the Atlantic ocean pout indicate the large size of its gene family

Type III AFP sequences from the Atlantic ocean pout (*Z. americanus*), the species in which this AFP was first discovered, were previously obtained by Edman degradation of proteins or from cDNA and genomic sequences [25, 26, 34]. Here, cDNA sequences encoding eight type III AFPs from this fish were cloned using 3´ RACE (S3 Table). To ensure that the sequence differences were not due to PCR or sequencing errors, only sequences represented by multiple clones are reported. There were between five (ocean pout-S2) and 25 (ocean pout-Q1) clones of each. The eight sequences were deposited in GenBank under accession numbers KR872957-KR872964, but throughout the manuscript, we refer to them by their corresponding protein accession numbers, ALL26673-ALL26680.

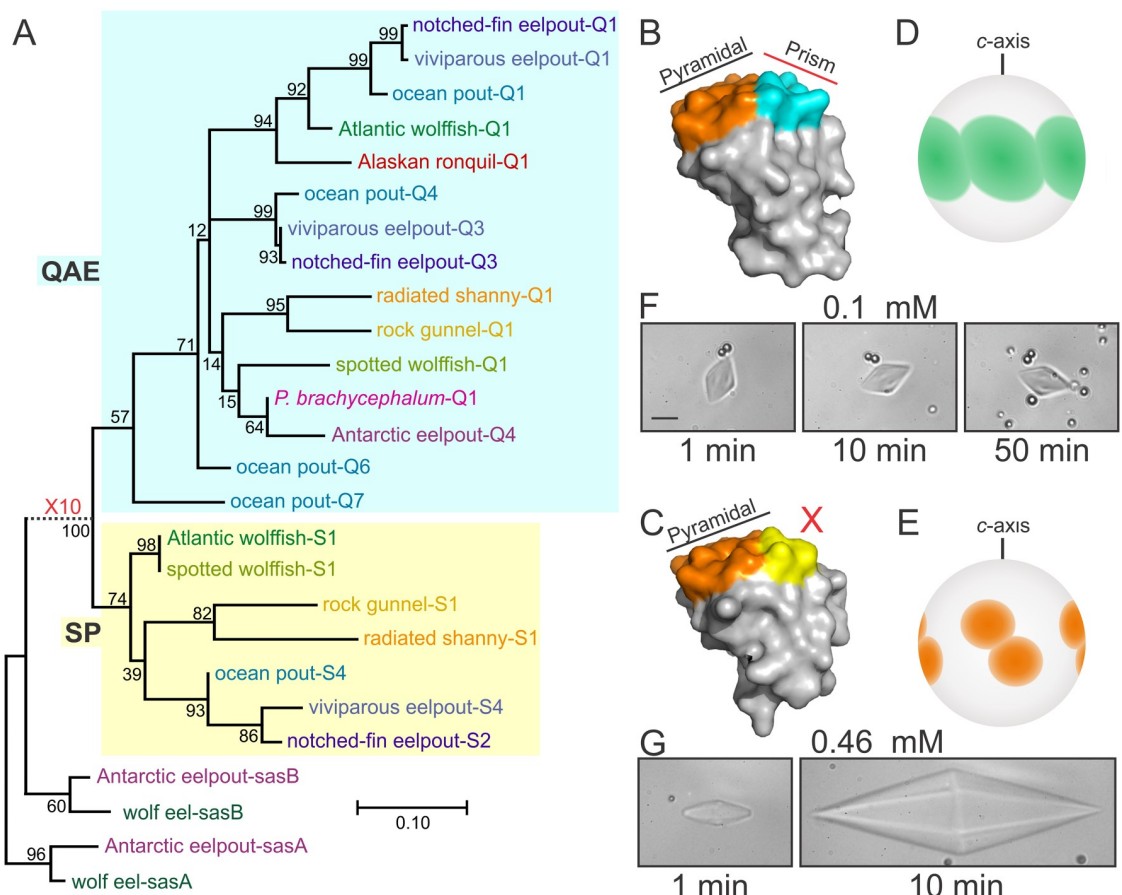

**Fig 3. Phylogenetic and functional comparison of type III AFPs.** A) A maximum-likelihood phylogenetic tree of the nucleotide sequences (S4 Fig) of the subset of type III AFP sequences shown in Fig 2. Cyan and yellow backing denotes QAE and SP isoforms respectively and bootstrap values (percent) are indicated at most nodes. The scale bar represents an average of 0.1 or 1 changes per site for solid and dashed lines respectively. B) and C) Representative structure of a QAE (PDB:4UR4) and SP isoform (PDB:4UR6) respectively [53] with the pyramidal and prism ice-binding surfaces colored orange and cyan respectively [78]. D) and E) Diagram of the fluorescent ice-plane affinity of a QAE and SP isoform respectively, adapted from previously published images [79]. F) and G) Ice crystals in the presence of 0.1 mM of a fully-active QAE isoform (M1.1, [91]) and 0.46 mM of an SP isoform (notched-fin eelpout-S5, S3 Fig [27]) respectively. Samples were cooled at a rate of 0.01˚C/6 sec for one min then held for the indicated times at 0.1˚C below the melting point. The scale bar represents 10 μm.

When the new ocean pout sequences were compared to known conspecific nucleotide and protein sequences (S3 Table), only two out of the eight matched previously known nucleotide sequences. Of these two, a sequence obtained from muscle (ALL26680) matches one previously obtained from pancreas (ocean pout-Q7, Fig 2, [34]), and a cDNA obtained from gill (ALL26678) matches the genomic clone OP5 [26]. Three others (ocean pout-Q5, ocean pout-S2 and ocean pout-S4) match isoforms known only from protein sequencing (HPLC12, HPLC7 and HPLC1, respectively) [26]. There is evidence of post-translational modification of HPLC1 and HPLC7 as the last two residues encoded by ocean pout-S2 and ocean pout-S4 (Gly-Lys) are absent. The Gly residue is likely acted on by peptidylglycine α-amidating monooxygenase [48] following removal of the Lys, generating an amidated C-terminal Ala. The three remaining sequences, ocean pout-Q1, ocean pout-Q4 and ocean pout-Q6 are unique. Two of these new sequences are SP isoforms (ocean pout-S2 and ocean pout-S4), three are QAE isoforms (ocean pout-Q1, ocean pout-Q4 and ocean pout-Q5) and the sixth (ocean pout-Q6) diverged early within the QAE lineage (Figs 2 and 3, S3 Fig).

## New variants from the transcriptomes of viviparous eelpout and *P. brachycephalum* strengthen the pattern of AFP relationships

Of the over 300,000 transcriptome sequence reads from the livers of eighteen viviparous eelpouts (*Z. viviparus*) from Scandinavian waters, some of which were from fish were harvested in November [49], 0.7% or ~2000 encoded AFPs. Therefore, this collection likely encompasses all of the sequences expressed in liver, but there may be tissue-specific genes expressed in other organs. A total of 19 unique sequences (12 SP, 7 QAE) were unambiguously assembled from these reads. One exactly matched a previously known protein sequence (AGM97733), while two others differed from ABN42204 and ABN42205 at two and three residues, respectively. Once sequences with two or fewer a.a. differences were excluded, 5 QAE and 8 SP sequences remained, designated viviparous eelpout-Q1 to -Q5 and -S3 to -S10 (S3 Fig). The pairs of SRA reads that can be used to generate these sequences are indicated in S4 Table.

The *P. brachycephalum* transcriptome sequence reads were generated from mRNA extracted from the hearts and livers of nine captured fish from Antarctica that were reared in tanks [50]. Only 165 reads, or 0.034% of over 480,000 reads obtained from normalized cDNA fractions, encoded AFPs. These assembled into groups encoding three distinct QAE isoforms (*P. brachycephalum*-Q1, -Q2, -Q4) with 75–89% identity, with *P. brachycephalum*-Q4 matching the previously-known cDNA sequence [34].

## Phylogenetic comparisons suggest extant type III AFP sequences arose only once

Type III AFP sequences with at least three amino acid differences (within a species) were aligned (Fig 2, S3 Fig) to trace the origin and relatedness of the different sequences both within and between species. Furthermore, the nucleotide alignment for the same sequences (S4 Fig) was considered the best choice for generating a phylogenetic tree for the following reasons. First, there are several informative silent-site mutations, in both the first and third codon positions. Second, some codons have two or three differences that are more informative than the single change represented by the amino acid. Third, the position of the gaps was easier to ascertain from the nucleotide alignment. Although the tree generated using the protein sequences was very similar (S7 Fig), the bootstrap values were lower and many more nodes were unresolved (polytomous). Nevertheless, the two AFPs known only from Edman degradation of purified protein (Canadian eelpout-Q1 and *P. brachycephalum*-Q3) clustered with the other sequences from the two Antarctic species.

The phylogenetic tree of a representative subset of the coding sequences (Fig 3) shows that most of the type III AFP sequences cluster into either the SP (yellow shading) or QAE (cyan shading) group. These two types were initially recognized in ocean pout [26]. The identity within the SP group ranges from 80–100%, with Atlantic wolffish-S1 and spotted wolffish-S1 being identical. Within the QAE group, identities are 73% or higher. Between the two groups, identities range from 66% to 89%. The similarity to the progenitor sequence (sasB) is lower, with identities ranging from 60 to 65%. The distance between the SAS cluster and the node at which the QAE and SP groups diverge is quite long, so it was shortened in Fig 3 (dashed line) for aesthetic reasons. This long branch as well as the near identity (96%) of the SAS-B C-terminal domains from wolf eel (*Anarrhichthys ocellatus*), a fish from the same family of northern fishes (Anarhichadidae) as the wolffish, and the Antarctic eelpout, indicate that the all of the AFP sequences are far more similar to each other than they are to SAS. This shows that the AFP arose from SAS one time only.

Two sequences from ocean pout (ocean pout-6 and ocean pout-7) appeared to be intermediate in nature, containing residues typical of both the SP or QAE groups, indicated by yellow

and cyan highlighting, respectively (Fig 2). As these variations are not contiguous, they are unlikely to have arisen by recombination or gene conversion. Rather, these alleles appear to have been duplicated soon after the QAE and SP gene lineages began to diverge, so they retain characteristics of both. There are a few other instances where residues typical of QAE sequences are found in SP sequences and vice versa, such as the Lys residue found at position 25 of some QAE sequences. Rather than being ancestral states, these appear to be reversions that occur subsequent to the initial mutations within each group.

The two *SAS* sequences from the wolf eel were obtained from a recent genome assembly from 150 bp paired-end Illumina reads (GenBank assembly accession GCA_004355925.1). These genes resided on a 5.7 Mb scaffold. Unfortunately, the four scaffolds containing AFP sequences (NW_022287273, NW_022287277, NW_022287306, and NW_022287306) were only ~2 kb in length and all four encoded an identical SP isoform. Due to the fragmentary nature of these assemblies, this *AFP* sequence, which was 96% identical to spotted wolffish-S2, was not analyzed further.

## SP sequences cluster along family lines

There are two main groups within the *SP* cluster (S8 Fig), in which sequences from Anarhichadidae (red dashed box) cluster separately from those within Zoarcinae (blue dashed box). The sequences from radiated shanny and rock gunnel also cluster. In contrast, the *QAE* sequences that are known show a much weaker association by family (Fig 4). This suggests that the common ancestor of these families may have possessed a larger number of *QAE* sequences than *SP* sequences. Alternatively, the *SP* genes may have undergone rounds of expansion and contraction more frequently than *QAE* genes.

## Viviparous eelpout sequences reveal recent AFP gene amplification

The amino acid sequence identity between the SP isoforms of the viviparous eelpout is at least 78% and between QAE isoforms it is at least 75%. This drops to between 55% and 66% between the two groups. All but one sequence (viviparous eelpout-Q5) closely clusters with sequences from the notched-fin eelpout (Fig 4). They also group closely with some of the ocean pout sequences, a result which is expected given the close relationship between these three species (Fig 1). Within the *SP* group, the ten viviparous eelpout and four notched-fin *SP* isoforms cluster separately from the four isoforms known from ocean pouts, albeit with some low bootstrap values (S8 Fig). This suggests that their genes have undergone multiple rounds of gene duplication and unequal crossing over (gene amplification) within the last few million years, after the ocean pout lineage separated from that leading to the notched-fin eelpout and viviparous eelpout lineages (Fig 1, SP↑). We should not be surprised at the plasticity of this and other *AFP* gene loci because there are other documented examples where *AFP* gene copy number is highly variable between closely related species [20] and even between the same species in different geographical regions [26].

## Antarctic AFP sequences cluster within a single QAE clade

Of interest in this study is the origin of *type III AFP* genes in the Antarctic zoarcids. Once highly similar sequences were eliminated, a total of 11 protein and 10 nucleotide sequences remained from the Antarctic species (S3 and S4 Figs). These cluster with high confidence into a single group, within the *QAE* clade (Fig 4). This pattern is consistent with *AFP* gene loss followed by reamplification from a single progenitor gene.

The encoded protein of one of the sequences recovered from the *P. brachycephalum* transcriptome exactly matched the known sequence (*P. brachycephalum*-4) [34, 41] but a few silent

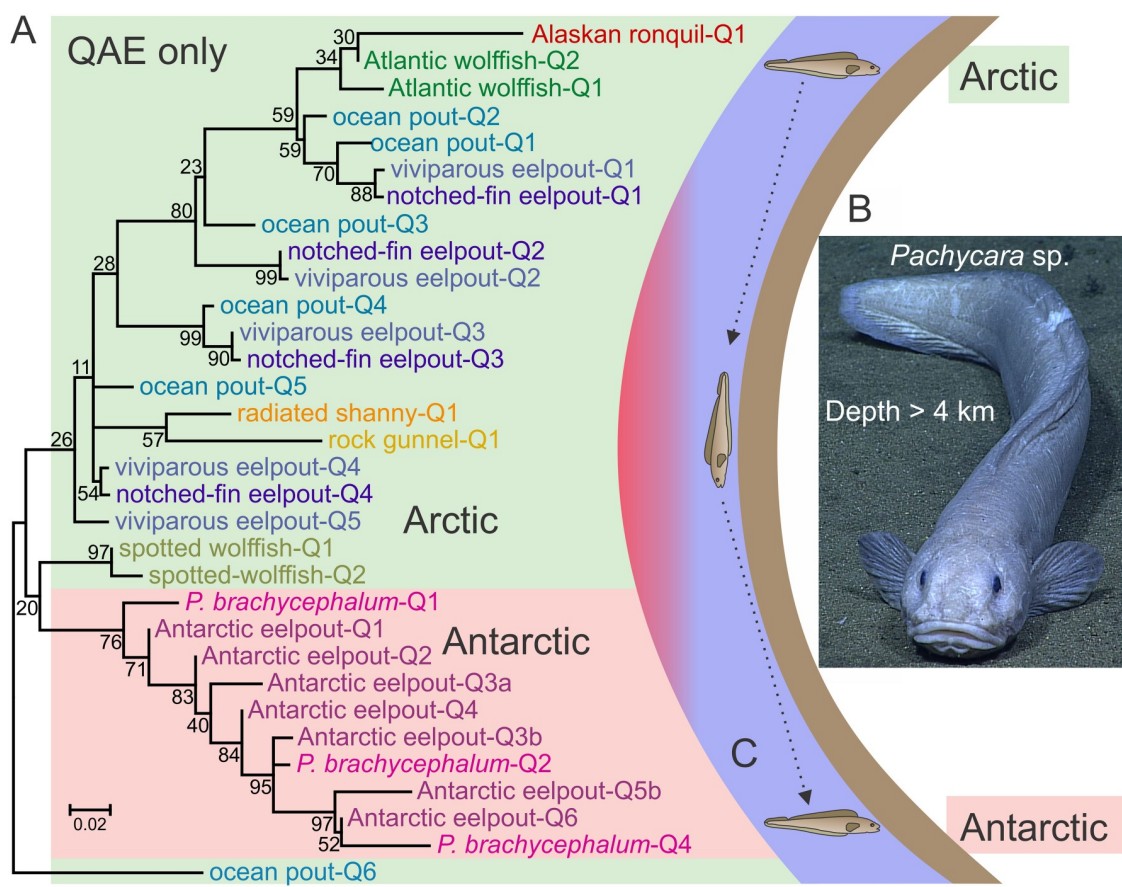

**Fig 4. Phylogenetic and geographical distribution of QAE isoforms.** A) Phylogenetic tree of the complete *QAE* subgroup from S4 Fig. Sequences are indicated as in Fig 2 and sequential tandemers from Antarctic eelpout are labelled alphabetically (e.g. Antarctic-eelpout-Q3a, Antarctic-eelpout-Q3b). Sequences from northern fish are on a green background while those from southern fish are on pink. The scale bar represents an average of 0.02 changes per site and bootstrap values (percent) are indicated at most nodes. B) A newly-discovered *Pachycara* sp., found during the 2016 NOAA Okeanos expedition at the Mariana Trench, 18°27′ N; 147°50′ E, in 1.5°C waters below 4000 m (https://service.ncddc.noaa.gov/rdn/oer-rov-cruises/ex1605l3/#tab-20). C) Diagram of a cold-water migration route through a cross-section of the oceans from the Arctic to the Anarctic. The deep water passage shown by the dotted arrows through the tropics avoids warmer surface waters (red).

or non-coding variations were also detected (not shown). The other two sequences, *P. brachycephalum*-Q1 and *P. brachycephalum*-Q2, were more similar (78 and 91% identity respectively) to the second sequence determined by protein sequencing (*P. brachycephalum*-3), which was isolated from fish living in a different area of Antarctica, approximately 4300 km distant [41, 50]. One of these (*P. brachycephalum*-Q1) is unique in having a 3′ UTR that does not match that of any known isoforms. The C terminus is similar to notched-fin eelpout-Q1 (CLCI vs CLCA, Fig 2), but this appears coincidental as this segment is also non-homologous given the Leu codon differs at two positions and the second Cys codon differs at the wobble position.

## Core and pyramidal ice-plane binding residues are well conserved

The protein alignments (Fig 2 and S3 Fig) show the variability at each position within the sequence. These differences have been mapped, using PyMol (1.7.6.3) [51], onto a stereo view of the highest resolution structure available (PDB 1UCS, [33]), which differs from Antarctic

eelpout-Q3a at one position (S9 Fig). Structurally important residues, such as the fifteen core residues, are highly conserved with conservative substitutions at only four positions (5-V/I/A, 22-M/I/T, 40-L/I/M, 55-L/I, S3 Fig). Two residues in turns, Pro29 and Pro33, are also conserved (S9A and S9B Fig). These are also the most highly-conserved residues relative to SAS-B (Fig 2), with conservation at all but two positions (22-M/L, 40-I/M), which underscores their importance in maintaining the hydrophobic core and overall structure of this domain/protein.

Residues found on the pyramidal ice-binding surface (IBS) are largely conserved (6 out of 9, green) with only three variable positions (13-I/M, light green, 15-T/S, light green (in only one isoform), 9-Q/V/T/R, grey) whereas the prism IBS is more variable, with only one out of five residues conserved (18-T, red), and four that are variable (orange and pale orange residues) (S9 Fig). One of the Antarctic QAE AFPs, *P. brachycephalum*-Q4, has residues on the IBS that are typical of SP isoforms (18–20, T**LV** to T**PA**, S3 Fig) and these significantly alter the prism IBS (S10A and S10B Fig). Conversely, some SP isoforms such as viviparous eelpout-S7 (S3 Fig) have T**LV** instead of T**PA**. When the most extreme substitution at each variable position with the IBS of all of the isoforms is mapped onto an ocean pout QAE structure [52], those on the prism IBP appear to change the surface flatness significantly (S10A and S10C Fig), whereas those on the pyramidal IBS (9, Q to R and 13, I to M) do not.

## Both SP and QAE isoforms are more hydrophobic than the progenitor

When the sequence of Antarctic eelpout SAS-B is mapped onto notched-fin eelpout QAE (PDB 4UR4) [53] using PyMol [51], the overall hydrophobicity of both the QAE (S10D Fig) and SP (PDB 4UR6) [53] (S10E Fig) isoforms relative to SAS (S10F Fig) is quite apparent. Here, a PyMol script that colors carbon atoms that lack hydrogen-bonding potential yellow and charges blue or red, was used [54]. The difference is most obvious on and flanking the IBS, as both the QAE and SP isoform are devoid of charged residues here. In contrast, SAS-B has three charged residues that protrude from this surface (K10, K13, D20) and three that are adjacent (K8, E47, D48), making this surface considerably more hydrophilic in SAS than in the AFPs.

The remaining surface residues of the AFPs are quite variable (S9 Fig, grey). There are many differences between QAE and SP isoforms (Fig 2, S3 Fig, yellow and cyan highlighting) or even within each group (Fig 2, S3 Fig, grey highlighting). However, there are ten of positions where QAE and SP isoforms share differences relative to SAS (no cyan or yellow, SAS highlighted, in black). The positions of two such residues, Thr54 and Asn46, are shown in S9 Fig. Their distance from the IBS would suggest they are not under selection, so these differences may have arisen by genetic drift following the duplication of the primordial AFP gene.

When the region encompassing the IBS is excluded, the AFPs are still more hydrophobic than the progenitor. Certain patches, such as residues 50–53 of QAE (PLGT) and SP (AKGQ), are significantly less charged relative to SAS (EEDD), but in some cases, such as position 23 of QAE, the reverse is true (S10D–S10F Fig). When the six charged residues near the IBS (above) and the last four residues of SAS-B are excluded, this domain still has an excess of six to seven charges over the AFPs. This is reflected in the relative percentage of solvent-exposed surface, calculated using PyMol [51], where it is lowest (57%) in the QAE isoform, slightly higher in the SP isoform (59%) and highest in SAS-B (64%) (S10D–S10F Fig). This is not only a property of these particular sequences alone, as the value for the portion of the sequence that was resolved in the NMR structure of human SAS (PDB 1WVO) [55] is similar to SAS-B (64%) and the QAE structure shown in S10A Fig (PDB 1HG7) [52] is similar to the SP isoform in S10E Fig as it has the same number of charged residues.

## Discussion

Type III AFP, which is found in fishes from a single branch of the taxonomic tree, may have allowed the Zoarcales to diversify and spread during the last 18 Ma, during a time of global cooling (Fig 1). Like most other fish AFPs, these are also present in multi-gene families [56]. We add to the body of evidence that this gene family has undergone many changes during this period, in which gene losses, gene duplications and other mutational events have occurred. The impetus for these changes is likely related both to the changing climate and to the migration patterns of the various species examined.

The Paleocene and early Eocene, from 65 to 45 Ma, was much warmer than today, with an ocean that was ice-free [18]. The presence of cold-intolerant tropical vegetation along the Antarctic coastline during the early Eocene [57] and deep ocean temperatures up to 14°C higher than today [18], support this assertion. The Zoarcales diverged from the Gasterosteiformes (sticklebacks and relatives) around this time [19], so the absence of type III AFPs outside of Zoarcales is not surprising, given the common ancestor of these two groups would have had no need for an AFP.

Once the Earth began to cool and ice was again present in the oceans, fish that acquired AFPs would be able to exploit "freeze-risk ecozones" with their abundance of invertebrate prey [29, 58]. This scenario is considered the impetus for the evolution and diversification of the four known AF(G)P types in diverse fish taxa [20]. An excellent example of ecozone exploitation is provided by the Notothenioids, which became the dominant taxon in Antarctic waters after acquiring AFGPs [59, 60].

The situation in the Northern Hemisphere is more complex as species with all four functional AFP types are found here [20]. In addition, the climate history of the Arctic is not as well understood. While Antarctic glaciation began ~35 Ma, the Arctic remained warmer for far longer with widespread glaciation apparently only occurring within the last 3.5 Ma [61, 62]. However, proxy evidence such as diatoms assemblages and ice-rafted debris suggest that ephemeral Arctic sea ice formed far earlier than previously thought, prior to 40 Ma [63, 64].

It is generally accepted that the infraorder Zoarcales originated in the Northern Pacific [65–68]. Therefore, our discovery of type III AFPs in two northern Zoarcales species that diverged over 10 Ma (radiated shanny, rock gunnel) and the Alaskan ronquil that diverged ~ 18.4 Ma (based on a time-calibrated phylogeny [42]), lends credence to the hypothesis that sea-ice was abundant in the North Pacific, well in advance of the opening of the Bering Strait [69–71] and many Ma before widespread glaciation occurred. It is likely that these groups remained in the Northern Pacific Ocean as nearly all species from the three families to which these fish belong (Bathymasteridae, Pholidae and Stichaeidae) still reside in this area [72]. It may be that the ancestors of the radiated shanny and rock gunnel migrated through the Bering Sea once the Bering Strait opened ~5 Ma [69–71], as the Bering Sea is thought to be the location where the Zoarcales underwent a major radiation [67]. The AFPs of these fish would have allowed them to survive in these icy northern waters. Species from these three groups never crossed the equator however [72], but as the surface waters in the tropical Pacific Ocean were only 1 to 2°C cooler during the last glacial maximum than they are today [73], they may have acted as a barrier to the spread of species adapted to survive in colder waters.

The SP and QAE types arose early within the Zoarcales lineage as they are found within all northern fishes for which multiple sequences were obtained [Zoarcidae [26], Anarhichadidae [29, 30], Pholidae and Stichaeidae (this study)]. The progenitor is under strong negative selection as the entire SAS sequence is highly conserved, as shown by the near-identity (96%) of SAS-B from a northern wolf eel (family Anarhichadidae) to that from Antarctic eelpout (family Zoarcidae). The AFPs are far more variable and show a high rate of mutation on surfaces

that are not involved in binding to the pyramidal ice plane. Despite this, all of the AFP sequences, whether they belong to the QAE or SP groups, are far more similar to each other than they are to SAS. Additionally type III AFP is only found in one infraorder (Zoarcales), whereas the three other AFP types that are known to have arisen by convergence or lateral gene transfer are found in fish from different orders [12, 14–17]. Taken together, it is very unlikely that type III AFP arose more than once from SAS (convergence, specifically parallelism). Instead, a single AFP likely arose early during the diversification of suborder Zoarcales and following one or more duplication events, gave rise to the SP and QAE isoforms that were transmitted by vertical descent to various families within Zoarcales as they arose.

As fishes from all but the most recently evolved family within Zoarcales are restricted to northern waters (Fishbase, [72]), the type III AFP clearly arose in the Northern Hemisphere. The eelpouts (Zoarcidae) originated around 10 Ma and are one of only a few families of fish found at both poles [74]. How then did some Zoarcids move from the far north to the Antarctic Ocean? It seems unlikely that cold-water fishes would migrate into warmer tropical waters and then back into cold Antarctic waters, all while retaining *AFPs*. They would need to first adapt to warmer water, where they would face competition from extant warm-water species, and then adapt back to colder water on the way to Antarctica. This would have been the case even during the coldest periods of the ice ages, because although the extent of the warmer waters was reduced, both the tropical Atlantic and Pacific waters were only a couple of degrees cooler than they are today [73]. However, there is another possible route, in both the Pacific and Atlantic basins, that does not require incursion into warm waters. Cold-adapted species could move from pole to pole through deep, perpetually cold waters (Fig 4). Indeed, Zoarcids are the predominant species near hydrothermal vents where the temperature a short distance away is typically around 2˚C, and *Pachycara* spp. have been found at depths exceeding 3 km in both the Atlantic and Pacific basins (Fig 4) [75, 76]. Interestingly, the majority of the Zoarcids that have been found near the equator are demersal (bottom dwelling) fish that have been recovered from depths of 500 m or more [72]. Unfortunately, the present-day distribution of the Antarctic eelpout and *P. brachycephalum* does not provide further clues as to their migration route as the Southern Ocean lacks land barriers.

Another possible scenario that could explain the presence of type III AFPs in southern Zoarcids is that the fish that migrated south did not have AFPs. Instead, they could have evolved anew from SAS once they encountered the icy Southern Ocean (parallelism). Work by Deng et al. (2010) in Antarctic eelpout showed that it was the SASb gene that was duplicated, with the 3´ exon encoding all but the signal peptide of the AFP [23]. The comparison between the C-terminal domain of SASa and SASb from wolf-eel, a northern fish within the same family as the spotted and Atlantic wolffishes, to the corresponding sequences from Antarctic eelpout, casts doubt on this hypothesis. If the AFPs arose independently from SAS progenitors in the two hemispheres (parallelism), the extant AFPs should show more similarity to their conspecific progenitors than to each other. However, the phylogenetic trees indicate the exact opposite (Fig 3). Additionally, there would be no reason for the Antarctic sequences to be most closely related to the single sequence known from Canadian eelpout, which is the closest relative to the Antarctic species in our study, with these two lineages having diverged less than 10 Ma (Fig 1). However, the Canadian eelpout sequence is 87% identical to Antarctic eelpout Q4 and clusters with the Antarctic sequences in the phylogenetic tree (S7 Fig). Furthermore, all of the *AFPs* clustered together near the end of a long branch (Fig 3), suggesting that the *AFPs* did not begin duplicating until the primordial gene had diverged significantly from SAS. Although it is plausible that the residues on the ice-binding surfaces could be similar due to convergence, the similarities between the variable surface residues away from these sites is

strongly suggestive of homology. Therefore, the AFPs within the Antarctic species are clearly homologs of those found in northern fishes.

An alternative hypothesis to parallelism is that the population that migrated south, long after the onset of Antarctic glaciation, may have lost all but one or a few nearly-identical *AFP* genes during the journey as all of the Antarctic sequences form a single cluster within the QAE portion of the phylogeny. While it is difficult to prove the complete absence of SP isoforms, the fact that none were recovered from serum or the transcriptome of *P. brachycephalum* [41, 50] or following screening of EST and BAC libraries from Antarctic eelpout [39, 40] strongly suggests they are absent. Unfortunately, genome sequencing has not proven useful for characterizing this multi-copy gene family, as shown by the failure of the *AFPs* of wolf eel to properly assemble. Therefore, the sequencing of individual BACs likely provides a clearer picture of the gene complement in Antarctic eelpout [23, 39]. Conversely, it is possible that other species of southern Zoarcids did retain SP isoforms and that there may have been more than one species that migrated into southern waters. The plasticity of the *AFP* gene family, both with regards to gene number and gene organization, has been clearly demonstrated. For example, the number of *AFP* genes, even within a single species, can change dramatically as was seen with ocean pout living at different latitudes [26]. Additionally, the organization can differ as the *AFP* genes are found in tandem repeats in ocean pout [26], as inverted pairs within tandem repeats in Atlantic wolffish [30] and in tandem repeats containing some tandemers at a single locus in Antarctic eelpout [23, 39]. rDNA genes are found in similar arrays and they are known to undergo rapid changes in copy number (reviewed in [77]). It should be noted that there is no well-documented advantage of AFPs to fish other than to protect them from internal ice growth. Thus, there is no need for AFPs in cold deep water where ice crystals are absent. Therefore, the phylogenetic analysis supports the hypothesis that gene duplication/amplification of the remaining *QAE type* gene(s) in the newly arrived migrants would have allowed these fish to survive in the icy Antarctic waters.

Further evidence for the plasticity of the *AFP* gene family is provided by the additional sequences we obtained from ocean pout, and by analyzing the transcriptome of the viviparous eelpout [49]. A variety of divergent QAE isoforms have been retained in both of these eelpouts as the QAE isoforms of these two species form a number of clusters. However, the SP isoforms of viviparous eelpout cluster together with those of the notched-fin eelpout but separately from those of ocean pout. This suggests that gene losses and duplications may have occurred frequently, particularly within the SP group, within the last 10 Ma. Still, both types have been retained. Many isoforms (all SP and some QAE) have been shown to impart hexagonal shaping to ice without preventing growth, due to structural differences in the ice binding site that prevents binding to the primary prism plane of ice [78, 79]. This has led some to suggest that these AFPs should be reclassified as "ice growth modifiers" [28, 80]. This limitation is overcome by the addition of minor quantities of a "fully active" QAE isoform [27, 28]. The isoforms appear to show synergism as the activity of the mixture is a little higher than that obtained with the QAE isoform alone [28]. Synergism has also been reported with AFGPs [81, 82] and beetle AFPs [83]. Therefore, the retention of multiple isoform types is likely advantageous for type III-producing fishes.

The two Antarctic species appear to have lost SP isoforms as mentioned above, as all of the isoforms from these two species cluster on a single branch within the QAE grouping of the gene phylogeny. The serum of *P. brachycephalum* isolated from McMurdo Sound was found to contain only two QAE isoforms [41] whereas transcriptome sequencing (over 480k sequences) from fishes isolated over 4000 km away in the South Shetland Islands [50] revealed three QAE isoforms, two of which were unique, but no SP isoforms. The isoform encoded by the most abundant transcript matched the predominant isoform isolated from the McMurdo

fishes. Interestingly, this isoform has two mutations that convert the ice-binding surface from QAE-like (T**LV**) to SP-like (T**PA**). Although it was reported to be fully active [41], this is no longer a certainty as we now know that trace amounts of fully-active QAE isoforms can confer full activity to SP isoforms [27, 78]. Why then would fish retain isoforms that are not fully active? One observation is that the SP-like mutations to the ice binding surface appear to extend the footprint of the pyramidal-plane binding surface while slightly reducing the overall hydrophobicity of the protein. This may strengthen or speed up ice binding while improving solubility of the AFP. Additionally, the AFPs are substantially more hydrophobic than Antarctic eelpout SAS-B, a difference that was also noted when the NMR structure of the corresponding domain of human SAS was solved [55]. While this invariably relates to function, as hydrophobicity is a general property of the IBS of type III AFPs and loss of the hydrophobic residues flanking the IBS can be deleterious [84], it may be that a mixture of QAE and SP isoforms is more soluble at high concentrations than one isoform alone. For example, a notched-fin eelpout SP isoform expressed in bacteria has shown a propensity to dimerize [53]. Furthermore, it seems unlikely that the ice-binding site of the dominant Antarctic QAE isoform would have mutated to an SP-like ice-binding site if there was no selective advantage to having both types.

The lack of an SP isoform in the Antarctic eelpout also seems probable, as sequencing of 61 AFP-encoding expressed sequence tags (ESTs) and six BAC clones containing multiple *AFP* genes [39, 40] failed to turn up a single SP isoform. The various isoforms from this species have a number of mutations on the ice-binding surface, such as TL**I** or TL**M** instead of TL**V**, but more importantly, this is the only species known to produce tandemers (sequential AFP domains encoded in a single transcript) [37, 39, 40]. This again suggests that fully-active QAE monomers alone might not be as effective as a mixture of isoforms and reveals yet another mechanism by which the diversity of the type III AFP family is attained.

It is tempting to try to deduce function from the sequences of the various isoforms of type III AFP, but unfortunately, the effect of the various mutations on AFP activity are very difficult to predict. Mutations of SP isoform notched-fin eelpout-S1, introduced to mimic the ice-binding residues of a fully-active QAE isoform (T**PA** to T**VL**), almost restored wild type activity. However, the crystals still grew, albeit at a much slower rate [78]. This may be because other residues are also know to affect activity. For example, a reduction in the size of hydrophobic residues bordering the ice-binding surface of a QAE isoform reduced activity [84]. Additionally, it is known that some QAE isoforms are not fully active [27, 28]. Therefore, it may be that the activities of closely-related sequences could vary and the only reliable way to determine this would be to express and purify these isoforms.

There are over 400 known species of Zoarcales [66], including others that inhabit the ocean near Antartica [67], so the sampling of 12 species catalogued in this study by us and others is by no means a comprehensive survey of all of the type III diversity that is present in this infra-order. There may well be northern species that lack SP isoforms or southern species that have retained them, although the two Antarctic species that have been characterized do appear to have lost their SP isoforms and duplicated/amplified their QAE isoforms. Lineage-specific amplifications of SP isoforms have also taken place in northern species. Additionally, we have shown that all of the known isoforms were derived from a single progenitor sequence and have further elucidated the variability of this gene family in this infraorder. The addition from this study of 24 sequences and their polymorphisms to the type III AFP family should also prove useful in modelling or mutagenesis studies aimed at further understanding the relationship between structure and function in type III AFPs, particularly as a number of these isoforms have mutations on or near the ice-binding surface that would be expected to alter their activity. In addition, we have shown that the type III AFP arose early in the Zoarcales lineage

and this may have been one of the factors that allowed this group of fishes to migrate into waters across the globe and to diversify over the last ~20 Ma into the over 400 species that are extant today.

## Materials and methods

### Sample collection and preparation

The Alaskan ronquil tissue sample preserved in ethanol was obtained from The University of Washington Fish Collection (Catalog number UW 150179). Ethanol was removed from the tissue and DNA was purified as described in Supplementary Methods.

Radiated shannys, rock gunnels and Atlantic ocean pout were collected near Logy Bay, Newfoundland, by divers from the Field Services Unit, Ocean Sciences Centre, Memorial University of Newfoundland, and transported in seawater to the Ocean Sciences Centre. Tissues were obtained from fish maintained at the centre that were euthanized by an overdose of MS222 just prior to tissue extraction. RNA was extracted from fresh or flash-frozen tissues as described in Supplementary Methods. Experiments were approved and carried out in accordance with Animal Utilization Protocols issued by Memorial University of Newfoundland's Animal Care Committee. All measures were taken to minimize pain and discomfort during animal experiments. Guidelines followed were those of the Canadian Council on Animal Care (CCAC).

### Type III AFP cloning and sequencing

Sequences that were obtained by PCR in this study are indicated by an asterisk while those assembled from the SRA database are indicated by a dagger (S3 Fig). A *type III AFP* sequence from the Alaskan ronquil was obtained from semi-nested PCR. Full-length cDNAs for *type III AFP* (QAE and SP isoforms) from radiated shanny and rock gunnel were cloned using a commercial kit for RNA ligase-mediated rapid amplification of 5´ and 3´ cDNA ends (RLM-RACE) [GeneRacer Kit (Invitrogen/Life Technologies)]. Partial cDNAs for *type III AFPs* from Atlantic ocean pout were cloned using the 3´ RACE protocol only. The sequences of all primers used in gene and cDNA cloning are presented in S1 Table. Detailed cloning and sequencing methods are provided in Supplementary Methods.

### Assembly of type III sequences from transcriptomes

Sequences from two transcriptomes, obtained from a northern viviparous eelpout ((*Z. viviparus*), SRA SRX002161) and southern Zoarcid (*P. brachycephalum* SRA SRX118640) were screened for sequences encoding AFPs. Longer sequences from the viviparous eelpout were selected to reduce the number of sequences that would exactly match more than one variant. These were grouped into sets of identical or near identical sequences, where the only allowed differences were length variations within homopolymeric runs. Sequences were assembled using the CAP3 Sequence Assembly Program [85]. Following assembly into nineteen (viviparous eelpout) or three (*P. brachycephalum*) unique sequences, reads that overlapped more than one of these unique sequences were discarded and the assemblies were redone.

### Phylogenetic analysis of predicted type III AFP protein sequences and structure rendering

The assembled Sequence Read Archive (SRA) data from above, along with type III AFP protein and nucleotide sequences from the protein and nucleotide databases of NCBI were aligned using Clustal Omega [86]. Sequences from the same species that differed at two or fewer

positions within the protein sequence (considering each AFP domain within the tandemers encoded in multiple BAC clones from Antarctic eelpout individually [23]) were considered redundant and were excluded. Minor adjustments were made to the protein alignment based on the nucleotide alignment. Phylogenetic trees were generated using nucleotide alignments and the SAS-B progenitor from Antarctic eelpout was used to root the tree containing both SP and QAE sequences [23] while a more divergent AFP sequence (ocean pout-7) was used to root the QAE tree. *MEGA* version 7 [87] was used to perform model tests prior to generating phylogenetic trees by the maximum likelihood statistical method using a moderate branch swap filter and all positions with 500 bootstrap replicates.

## Supporting information

**S1 Fig. Alignment of cytochrome oxidase subunit I (*COI*) sequences from various members of the infraorder Zoarcales.** GenBank accession #s, sequentially top to bottom as in the alignment, are KC016052, KJ205263, KC517318, HQ712639, HQ713113, HQ713057, KJ205118, KC016016, JQ685890, KC015305, EU752057, EF427917. Since a *COI* sequence was not available for the Antarctic eelpout, *Lycodichthys dearborni*, the one for *Lycodichthys antarcticus* was used instead. Asterisks under the alignment indicate perfect conservation of a base.
(DOCX)

**S2 Fig. Phylogenetic tree of Zoarcales species using the *COI* alignment from S1 Fig.** The maximum-likelihood tree was generated using the Kimura 2-parameter with invariant sites model and bootstrap values (%) are indicated at the nodes. The Alaskan ronquil was used as the outgroup and the scale bar represents an average of 0.02 changes per site. This tree was used to determine the placement of the radiated shanny, Antarctic eelpout and *Pachycara brachycephalum* relative to the other species in Fig 1.
(DOCX)

**S3 Fig. Protein alignment of known type III AFPs that have more than two differences relative to other sequences from the same species.** Sequences are named as in Fig 2. Grey highlighting indicates sequences that were determined by Edman degradation. The tandemers of Antarctic eelpout are lettered sequentially (e.g. Antarctic eelpout-Q3b is the second AFP domain in Antarctic eelpout-Q3). Variable residues are highlighted or coloured according to the phylogenetic tree (Fig 3) with conserved residues typical of QAE or SP variants highlighted cyan and yellow, respectively. Mutations that arose somewhere within Antarctic species are highlighted pink and residues within SAS-B that were not conserved when the QAE or SP groups arose are highlighted black. Other differences between SAS sequences are highlighted purple. Differences that do not correlate with these aforementioned groupings are highlighted grey. Red highlighting indicates shared differences between the signal peptides of the sequences from radiated shanny and rock gunnel. The black boxes and red boxes show residues involved in binding to the pyramidal plane and prism plane respectively, as in Fig 2. The signal peptide is in lowercase font. Italics indicate linkers between tandemers, internal dashes indicate gaps, whereas leading or trailing dashes indicate that the sequence is incomplete at either terminus. Identity, high similarity and low similarity between all AFPs (incomplete sequences ignored) is indicated at the bottom with asterisks, colons and periods, respectively. Residues with an inward pointing sidechain are indicated by "i" at the top. Asterisks denote sequences obtained by PCR in this study (S3 Table) and daggers denote those assembled from the SRA database (S4 Table). Accession numbers are listed in S4 and S5 Tables.
(PDF)

**S4 Fig. Nucleotide alignment of type III AFPs.** Sequences are named and variable nucleotides are highlighted as in Fig 2. Red highlighting indicates shared differences between the first exons of radiated shanny and rock gunnel and these exons were excluded prior to generating all but the exon 1 phylogenetic tree as they may have been homogenized by exon shuffling. The translation of notched-fin eelpout-Q1 has black boxes and red boxes showing residues involved in binding to the pyramidal plane and prism plane respectively, as in Fig 2. The signal peptide is in lowercase font. Internal dashes indicate gaps, whereas leading or trailing dashes indicate that the sequence is incomplete at the respective terminus. Intronic sequences are not shown for genomic clones but an arrow indicates where an intron is found. The 3′ splice junction of ocean pout-Q2 was originally predicted based on SP isoforms but has been adjusted by three bases (lower-case font) to match QAE-type cDNAs. The 3′ end of *P. brachycephalum*-Q1 (italics) and the linker sequences between Antarctic eelpout tandemers (not shown) were excluded from the phylogenetic analysis as they are not homologous to the other AFP sequences. These nucleotide sequences are unambiguously accessed through the protein accession numbers in S4 and S5 Tables as some of the nucleotide sequences encode multiple AFPs.
(DOCX)

**S5 Fig. Maximum-likelihood phylogenetic tree of all non-identical exon 1 sequences from type III AFPs from S4 Fig generated using the Jukes-Cantor model with 5 gamma categories.** Cyan and yellow backing denotes QAE and SP isoforms, respectively, except for radiated shanny and rock gunnel sequences that cluster together on a separate branch. Bootstrap values (percent) are indicated at most nodes and the scale bar represents an average of 0.1 changes per site. Sequences are named as in Fig 2.
(TIF)

**S6 Fig. Tissue distribution of rock gunnel and radiated shanny type III AFPs.** Northern blot analysis of total RNA from two rock gunnel individuals (A and B) and two radiated shanny individuals (C and D). The panels in each set show the hybridization signal to the *AFP* probe (top); the chicken β-tubulin probe (middle) cDNAs; or ethidium bromide staining of the 28S and 18S rRNA bands (bottom). The tissues are indicated as follows; L = liver, Sk = skin, G = gill, I = intestine, St = stomach, M = muscle, H = heart, Sp = spleen, K = kidney. RNA size marker positions are indicated on the left (bases) and total RNA from cunner skin was used as a negative control (n).
(TIF)

**S7 Fig. Maximum-likelihood phylogenetic tree generated from protein sequences.** The amino acid sequences shown in Fig 2, along with two sequences determined solely by Edman degradation of purified proteins (Canadian eelpout-Q1 and P. brachycephalum-Q3, labelled with #) were used to generate a phylogenetic tree equivalent to Fig 3.
(TIF)

**S8 Fig. Phylogenetic comparison of SP isoforms.** The nucleotide sequences of the SP subset of *type III AFP* sequences (S4 Fig) were used to generate a maximum-likelihood phylogenetic tree using a divergent isoform (ocean pout-Q6) as the outgroup. Bootstrap values (percent) are indicated at most nodes and the scale bar represents an average of 0.02 changes per site. Sequences are named as in Fig 2.
(TIF)

**S9 Fig. Stereoscopic views of type III AFP showing the location of variable and conserved residues.** Residues that are absolutely, moderately or poorly conserved are colored as follows: respectively; pyramidal ice-binding plane, dark green, light green, yellow; prism ice-binding

plane, red, orange, pale orange; rest of protein, dark blue, light blue, grey. Front view (A), back view (B) and front surface view (C).
(TIF)

**S10 Fig. Surface view showing the expected structural effect of ice-binding residue mutations and the hydrophobicity of different isoforms.** A) Wild-type QAE isoform, ocean pout-Q5 (HPLC12, PDB 1HG7) B) Ocean pout-Q5 with the introduction of the three ice-binding mutations found in P. brachycephalum-Q4 and C) A compilation of the most severe mutations at variable ice-binding residues (S3 Fig) introduced to ocean pout-Q5. Nitrogen is blue, oxygen is red, sulfur is yellow and carbon is pale orange (on the pyramidal-plane ice-binding surface), cyan (on the prism-plane ice-binding surface) or white (elsewhere). D) QAE isoform (PDB 4UR4) E) SP isoform (PDB 4UR6) F) Antarctic eelpout SAS-B residues mapped onto 4UR4 (excluding the last four residues of SAS-B). Atoms are colored by charge and hydrophobicity with red for charged oxygen, blue for charged nitrogen and yellow for carbon not bonded to nitrogen or oxygen. All other backbone and polar groups are colored white. Residues are numbered according to Fig 2, except residues 50–53 (PLGT, AKGQ and EEDD respectively). The percentage of the surface that is solvent accessible is indicated.
(TIF)

**S1 Table. Sequences of oligonucleotides used in PCR studies.** *F is forward and R is reverse direction.
(DOCX)

**S2 Table. Rock gunnel and radiated shanny type III AFP cDNA and predicted protein features (from ProtParam [92]).** [1]Excludes poly(A) tail. [2]Includes STOP codon. [3]Excludes STOP codon and poly(A) tail. [4]Presumes cleavage of C-terminal Lys.
(DOCX)

**S3 Table. Ocean pout sequences cloned in this study and their closest nucleotide and protein matches.** SP and QAE isoforms are highlighted yellow and cyan, respectively, with the QAE sequences that diverged early are highlighted grey. Protein accession numbers are used for consistency between figures. Percent identity excludes gaps, and sequences known only from Edman degradation are underlined. Sequences denoted with an asterisk were not included in the alignments as they differed at two or fewer a.a. resides from isoforms that were included.
(DOCX)

**S4 Table. Accession numbers for one or two sequences from the NCBI SRA database that will generate viviparous eelpout (SRA accession # SRX002161) and *P. brachycephalus* (SRA accession # SRX118640) isoforms as shown in Fig 2 and S3 Fig.**
(DOCX)

**S5 Table. Genbank protein accession numbers for sequences shown in Fig 2 and S3 and S4 Figs.**
(DOCX)

**S1 File. Supplement for materials and methods, results, and references.**
(DOCX)

**S1 Raw image.**
(PDF)

## Author Contributions

**Conceptualization:** Rod S. Hobbs, Garth L. Fletcher.

**Data curation:** Rod S. Hobbs, Jennifer R. Hall, Laurie A. Graham, Garth L. Fletcher.

**Funding acquisition:** Peter L. Davies, Garth L. Fletcher.

**Investigation:** Rod S. Hobbs, Jennifer R. Hall, Laurie A. Graham.

**Project administration:** Peter L. Davies, Garth L. Fletcher.

**Resources:** Peter L. Davies, Garth L. Fletcher.

**Supervision:** Peter L. Davies, Garth L. Fletcher.

**Validation:** Rod S. Hobbs, Jennifer R. Hall, Laurie A. Graham.

**Visualization:** Rod S. Hobbs, Jennifer R. Hall, Laurie A. Graham.

**Writing – original draft:** Rod S. Hobbs, Jennifer R. Hall, Laurie A. Graham.

**Writing – review & editing:** Rod S. Hobbs, Jennifer R. Hall, Laurie A. Graham, Peter L. Davies, Garth L. Fletcher.

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
