## [Decision Letter · Decision Letter 0]

4 Mar 2020

PONE-D-19-34926

Antifreeze protein dispersion in eelpouts and related fishes reveal migration and climate alteration within the last 20 Ma

PLOS ONE

Dear Dr. Davies,

Thank you for submitting your manuscript to PLOS ONE. After careful consideration, we feel that it has merit but does not fully meet PLOS ONE’s publication criteria as it currently stands. Therefore, we invite you to submit a revised version of the manuscript that comprehensively addresses the points raised during the review process.

We would appreciate receiving your revised manuscript by Apr 18 2020 11:59PM. To enhance the reproducibility of your results, we recommend that if applicable you deposit your laboratory protocols in protocols.io, where a protocol can be assigned its own identifier (DOI) such that it can be cited independently in the future. For instructions see: http://journals.plos.org/plosone/s/submission-guidelines#loc-laboratory-protocols

We look forward to receiving your revised manuscript.

Kind regards,

Michael Schubert

Academic Editor

PLOS ONE

PLOS ONE now requires that submissions reporting blots or gels include original, uncropped blot/gel image data as a supplement or in a public repository. This is in addition to complying with the image preparation guidelines described at https://journals.plos.org/plosone/s/figures#loc-blot-and-gel-reporting-requirements . These requirements apply both to the main figures and to cropped blot/gel images included in Supporting Information. 

Journal Requirements:

3. Thank you for including your ethics statement:  "Tissues were obtained from fish maintained at the Ocean Sciences Centre, Memorial University of Newfoundland that were euthanized by an overdose of MS222 just prior to tissue extraction. Experiments were carried out in accordance with Animal Utilization Protocols issued by Memorial University of Newfoundland’s Animal Care Committee. All measures were taken to minimize pain and discomfort during animal experiments. Guidelines followed were those of the Canadian Council on Animal Care (CCAC)."

Please amend your current ethics statement to confirm that your named institutional review board or ethics committee specifically approved this study.

For additional information about PLOS ONE submissions requirements for ethics oversight of animal work, please refer to http://journals.plos.org/plosone/s/submission-guidelines#loc-animal-research  

4. Please upload a copy of Supporting Information Figure S4 which you refer to in your text on page 10.

5. Please upload a copy of Supporting Information Table S1-S4 which you refer to in your text on page 7, 9 and 21.

Reviewers' comments:

Reviewer's Responses to Questions

**Comments to the Author**

1. Is the manuscript technically sound, and do the data support the conclusions?

Reviewer #1: Partly

Reviewer #2: No

2. Has the statistical analysis been performed appropriately and rigorously? 

Reviewer #1: Yes

Reviewer #2: No

3. Have the authors made all data underlying the findings in their manuscript fully available?

Reviewer #1: Yes

Reviewer #2: No

4. Is the manuscript presented in an intelligible fashion and written in standard English?

Reviewer #1: Yes

Reviewer #2: Yes

5. Review Comments to the Author

Reviewer #1: Dear Dr. Hobbs et al.,

I have now read through the submitted ms PONE-D-19-34926 in detail – and I do have some concerns that I will specify in detail below:

First, I have some issues with the readability of the paper, i.e. which question that was asked vs how the results were presented. F. ex. I would have loved to see more readable figures that includes the full common name and/or scientific names. As presented now (from Figure 2 and onwards) the authors have only listed the initials of the common names (then I needed to double check the actual species named over and over again (a bit frustrating and very confusing for people that are not that familiar with the group of fishes that have been studied here).

And: not sure if which tissue needs to be specified in the figure? Think the different sequences could be denoted a number – and linked to tissue f. ex. in a supp table (since which tissue the sequence is obtained from seems not to be in focus in this paper).

Further, it is not clear which data the authors have generated themselves vs. data/sequences that they have collected from databases (GeneBank ++). Maybe this could have been elaborated in the very beginning of the Mat and Met/Results section. And by such specify the samples collected/investigated that were needed to fill the gaps/answered the specific questions in mind? This is somewhat touched upon in the intro – but could preferably be elaborated more in the Mat and Met section.

In this regard I would also suggest some re-writing of the introduction – to get the message through to the reader. In the intro you start out with a general paragraph re the antifreeze proteins in fish – which is great. Then in the second paragraph moving into the details of AFP III which is the main focus of the ms – which is also OK as is. The next (third) paragraph lists the other AFPs and AFGPs and how they have evolved and then and a fourth paragraph about the timing of the divergence of the different lineages that produced AFP. I would suggest that these two paragraphs are combined into one – and maybe focus on the evolutionary aspects (convergent evolution and timing the most; a bit descriptive as is). And this paragraph could preferably have become the second paragraph of the intro – setting the stage of the question in mind. And then going into the detail about the AFP type III afterwards and then moving into the questions to be asked within this lineage. I would then also have moved upward the description about the early southern blotting studies up (line 95-101) + the description about the Antarctic species (104-112) to the AFP type III paragraph (which I suggest to be the third paragraph). Can maybe be divided in to two paragraphs. But then at least, you could list the specific scientific questions in on go at the very end of the introduction. As is, you give some introduction/background and list the question, move on to some more details/background information and a new question. Think it would be better to combine the background information and then list the questions in one go at the very end.

How the results are presented is also a bit messy. Would have first given the reader the results of the phylogeny of this infra order (confirming the Figure 1 which is presented in the ms). Then I would have presented the phylogenetic gene trees of the APF sequences together with the full-length cDNA sequences encoding both QAE and SP isoforms from rock

gunnel and radiated shanny as well as the additional sequences identified for the P. brachycephalum and the Atlantic ocean pout (and for the two latter some of the detailed description could go into the Mat and Met or supp I think).

And one last comment re the result section – do not think the subtitles are optimal should be re-written to mirror the results presented in a better way.

Then to the inference of the data, where I think the conclusions about the timing of the evolution of the AFPs within this lineage and that QAE sequences origin from direct decent are solid. However, I do question if they could say the same for the SPs? Here, we do see a lager degree of homology within families (that they cluster together) which is not the case for the QAE sequences (at the same degree at least).

I would say that the higher divergence between species for the QAE sequences indicate strongly that they origin from the same ancestor QAE sequence, that have then evolved in the different lineages in different paces/slightly different directions. But for SP the results could indicate that the SP sequences have evolved over and over again independently for the different families (from the SAS sequence specific for the different species and/or the common ancestor for the specific families). Think this cannot be ruled out – and maybe also plausible – since the SPs is only functional in combination with the QAE type (and could be looked upon as added value in addition to the QAE under certain circumstances I guess). Think this or higher degree of divergent selection could explain this (more than gene losses and duplications as the authors state). This could and should be further looked into by performing dN/dS analyses of the sequence data. Would also strongly suggest to do these analyses for the QAE sequences too.

And f. ex not sure if I truly agree on the statement re the QAE sequences on line 255-256: “This suggests that the common ancestor of these families may have possessed a larger number of QAE sequences than SP sequences.”

These speculations, as well as about more gene losses and duplications being responsible for the clustering of the SP sequences can only be inferred if they have had whole genome sequencing data. Same goes for the plausible loss of the SP in the Antarctica species. Full genome data set is needed to confirm a loss) so should be careful here I think.

Additionally, in this paper only two Antarctic species are investigated (it is listed that this linage contain about 100 species so I guess some of them could have the SP type (i.e. not yet confirmed but then either retained if lost in some of them and/or evolved from the SAS sequence and/or the QAE sequence in the common ancestor of this lineage)).

And then what I really miss in this ms is the sequence data for SP for the Canadian eelpout, would have been beneficial to have that one to see how divergent this SP sequence is compared to the other ones. Any change that the authors could get hold of that sequence information? This information would for sure enlighten the evolutionary path of the SP within this lineage.

My last comment is about the speculation how the eelpouts have migrated down south. Is there not a possibility that the eelpouts have moved northwards and southwards from different refugia? And diversified in different rounds after settlement in the different regions? Would have loved to see some more speculations here – and then not at least linked more to past paleoclimatic events during the Miocene (cooling ++ as well as different possible refugia during that period). Think this explanation is more likely to have found place.

In relation to this, they write that the eelpouts might have migrated via the cold depths and at the same time (on the way down) lost their SP due to lack of ice crystals at these depths. Not sure if the “red” color on Figure 4 is describing this in a good way as well as the migratory route (not that illustrative). Would suggest to modify this figure and also take into account the other possibilities re coming for different refugia f. ex.

One minor comment:

Would strongly suggest to use gene duplication not amplification.

Reviewer #2: Antifreeze proteins are fascinating examples of convergent evolution of function. Four different types of antifreeze proteins have evolved in fish species throughout the globe. This manuscript addresses three questions about the evolution of Type III antifreeze proteins in Zoarcales including whether the proteins arose by direct descent vs parallel evolution, when the antifreeze proteins (AFPs) arose, and the timing of the colonization of the Antarctic by zoarcids.

The finding of the AFP in the Alaskan ronquil is exciting and challenges the current accepted timeline for the evolution of AFPs. I agree with the manuscript that it also provides additional evidence for an earlier cooling of the Arctic Ocean.

The evidence for the evolution of QAE and SP isoforms is contradictory. While the phylogenetic tree places the appearance of the QAE at the Bathymasteridae (Figure 1), the tree of relationships in Figure 3 suggests that the SP isoform is ancestral and the QAE isoform is the derived state, especially with regards to the placement of the OPpan and the AEsasB. To simply state it: Figure 1 shows QAE arose first and SP is derived whereas Figure 3 shows the opposite (SP is ancestral and QAE is derived). The results in Figure 4 are also contradictory with the statements regarding the Antarctic having the highly derived sequence. Many of the branches have little to no support, which suggests the branches should be collapsed and not interpreted as evidence for the findings. Moreover, the authors propose that “likely lost all but one or a few nearly-identical AFP genes during the journey [from the Arctic to Antarctic]” and that “gene losses and duplications have occurred frequently, particularly within the SP group” so it is possible that SP were lost in other lineages and the SP form arose first.

The manuscript highlights a recent gene amplification in the Viviparous eelpout, however, there is no discussion of the L. dearborni expansion. There is data available from ~30 L. dearborni AFPs on GenBank through BAC sequencing (Deng et al 2010), which has the genomic context for the expansion.

The findings are generally confounded by the duplications and deletions that are presumably occurring very frequently.

For some of the sections, the authors provide alternative explanations. The alternative explanations are incredibly helpful in interpreting the results and I appreciate the inclusion.

The figure legends and text are confounded and it is hard to determine what is in the figure legend and what is main text.

Initials of a common name seems to be an unusual way to label sequences in the text and Figure 2 (sequence alignment).

The tissue of origin is included on some of the isoforms, does the tissue of origin matter? The liver is the most common place for transcription of AFPs but there seem to be tissue-specific transcripts. Are those relevant for the organismal performance? There are several similar smaller results in the manuscript where it is unclear the relevance to the larger findings.

None of the supplementary tables are available.

Figure S4 is not referred to in the main text and is not available in the supplementary materials.

Is there evidence for loss of the SP isoform?

Figure 4C is not informative.

There are some references that are missing (for example, on line 111).

Line 269 has some specific PLOS guidelines that should be removed.

6. PLOS authors have the option to publish the peer review history of their article (what does this mean?). If published, this will include your full peer review and any attached files.

Reviewer #1: No

Reviewer #2: No

---

## [Author Response · Author response to Decision Letter 0]

21 Apr 2020

Dear Dr. Schubert

Thank you for coordinating the review of our manuscript and for inviting us to submit a revision that addresses the criticisms raised by the reviewers. We apologize for the lack of inclusion of the supplementary tables and the figure legends during the submission process. We have addressed our responses to the reviewers point by point as documented below.

Reviewer Comments:

Reviewer 1

I have now read through the submitted ms PONE-D-19-34926 in detail – and I do have some concerns that I will specify in detail below:

First, I have some issues with the readability of the paper, i.e. which question that was asked vs how the results were presented. F. ex. I would have loved to see more readable figures that includes the full common name and/or scientific names. As presented now (from Figure 2 and onwards) the authors have only listed the initials of the common names (then I needed to double check the actual species named over and over again (a bit frustrating and very confusing for people that are not that familiar with the group of fishes that have been studied here).

- We have taken this advice and have altered all the figures and tables to include the full common name, or a slight contraction, where necessary. In addition, we have removed the accession numbers from figure 2 and the supplementary figures and inserted them into the corresponding figure legends.

And: not sure if which tissue needs to be specified in the figure? Think the different sequences could be denoted a number – and linked to tissue f. ex. in a supp table (since which tissue the sequence is obtained from seems not to be in focus in this paper).

- This point was also raised by Reviewer 2, and as described below information about tissue origin has been removed from the figures. However, this information can still be accessed from Table S3.

Further, it is not clear which data the authors have generated themselves vs. data/sequences that they have collected from databases (GeneBank ++). Maybe this could have been elaborated in the very beginning of the Mat and Met/Results section. And by such specify the samples collected/investigated that were needed to fill the gaps/answered the specific questions in mind? This is somewhat touched upon in the intro – but could preferably be elaborated more in the Mat and Met section.

- We have now distinguished the data we generated ourselves from those that came from databases by using a double underline in sequence alignments and in phylogenetic trees to denote sequences we obtained via PCR, and a single underline for those newly assembled from the sequence read archive in GenBank. This is now stated in Materials and Methods at line 422 and in the legend to Fig 2. No underlining is used for sequences obtained from the protein or nucleotide databases of GenBank. 

In this regard I would also suggest some re-writing of the introduction – to get the message through to the reader. In the intro you start out with a general paragraph re the antifreeze proteins in fish – which is great. Then in the second paragraph moving into the details of AFP III which is the main focus of the ms – which is also OK as is. The next (third) paragraph lists the other AFPs and AFGPs and how they have evolved and then and a fourth paragraph about the timing of the divergence of the different lineages that produced AFP. I would suggest that these two paragraphs are combined into one – and maybe focus on the evolutionary aspects (convergent evolution and timing the most; a bit descriptive as is). And this paragraph could preferably have become the second paragraph of the intro – setting the stage of the question in mind. And then going into the detail about the AFP type III afterwards and then moving into the questions to be asked within this lineage. I would then also have moved upward the description about the early southern blotting studies up (line 95-101) + the description about the Antarctic species (104-112) to the AFP type III paragraph (which I suggest to be the third paragraph). Can maybe be divided in to two paragraphs. But then at least, you could list the specific scientific questions in on go at the very end of the introduction. As is, you give some introduction/background and list the question, move on to some more details/background information and a new question. Think it would be better to combine the background information and then list the questions in one go at the very end.

- The introduction has been extensively edited and rearranged to follow these suggestions and address the reviewer’s concerns. These changes can be seen in the marked-up version of the manuscript. 

How the results are presented is also a bit messy. Would have first given the reader the results of the phylogeny of this infra order (confirming the Figure 1 which is presented in the ms). 

- The phylogeny results have been moved forward so the alignment and tree that support Figure 1 are now the first and second Supplementary Figures.

Then I would have presented the phylogenetic gene trees of the APF sequences together with the full-length cDNA sequences encoding both QAE and SP isoforms from rock

gunnel and radiated shanny as well as the additional sequences identified for the P. brachycephalum and the Atlantic ocean pout (and for the two latter some of the detailed description could go into the Mat and Met or supp I think).

- Done

And one last comment re the result section – do not think the subtitles are optimal should be re-written to mirror the results presented in a better way.

- We have revised many of the sub-titles to make them more descriptive of the results being presented in the following text.

Then to the inference of the data, where I think the conclusions about the timing of the evolution of the AFPs within this lineage and that QAE sequences origin from direct decent are solid. However, I do question if they could say the same for the SPs? Here, we do see a lager degree of homology within families (that they cluster together) which is not the case for the QAE sequences (at the same degree at least).

I would say that the higher divergence between species for the QAE sequences indicate strongly that they origin from the same ancestor QAE sequence, that have then evolved in the different lineages in different paces/slightly different directions. But for SP the results could indicate that the SP sequences have evolved over and over again independently for the different families (from the SAS sequence specific for the different species and/or the common ancestor for the specific families). Think this cannot be ruled out – and maybe also plausible – since the SPs is only functional in combination with the QAE type (and could be looked upon as added value in addition to the QAE under certain circumstances I guess). Think this or higher degree of divergent selection could explain this (more than gene losses and duplications as the authors state). This could and should be further looked into by performing dN/dS analyses of the sequence data. Would also strongly suggest to do these analyses for the QAE sequences too.

- The sequence differences between the SAS C-terminal domain and SP-type AFP isoforms are considerable and much more extensive than between SP sequences. Therefore, it seems unlikely that such similar SP sequences would independently evolve over and over again. Nevertheless, as a theoretical possibility we have included this scenario at your suggestion. 

- We have performed the suggested dN/dS analyses, which suggest the AFP sequences are under positive selection. We describe these results beginning on line 276 and show the data in Fig S10A.

And f. ex not sure if I truly agree on the statement re the QAE sequences on line 255-256: “This suggests that the common ancestor of these families may have possessed a larger number of QAE sequences than SP sequences.”

- We do realize that this statement is speculative and as these genes have likely undergone many rounds of expansion and contraction, but the greater variability within the QAE group does indicate that it is a possibility.

These speculations, as well as about more gene losses and duplications being responsible for the clustering of the SP sequences can only be inferred if they have had whole genome sequencing data. Same goes for the plausible loss of the SP in the Antarctica species. Full genome data set is needed to confirm a loss) so should be careful here I think.

- Your point is well taken. It is worth noting that genome sequencing is the easy part. Assembling and annotating the genome is incredibly difficult and very few have been done properly from start to finish. A good example where genome sequencing of an AFP-producing fish was not helpful is described in Zhuang X, Yang C, Fevolden SE, Cheng CH. Protein genes in repetitive sequence-antifreeze glycoproteins in Atlantic cod genome. BMC Genomics. 2012 13:293. doi: 10.1186/1471-2164-13-293. PMID: 22747999. There, the large number AFGP genes of a northern cod were missed in the genome assembly due to their repetitive nature.

Additionally, in this paper only two Antarctic species are investigated (it is listed that this linage contain about 100 species so I guess some of them could have the SP type (i.e. not yet confirmed but then either retained if lost in some of them and/or evolved from the SAS sequence and/or the QAE sequence in the common ancestor of this lineage)).

- If any species had an AFP that had convergently evolved (parallelism) from the SAS sequence, it should form a distinct clade within the phylogenetic tree in Figure 3. What we observe is a very long branch to SAS (dotted line) and a clustering of all known type III AFPs, including those from these two Antarctic species. This point is made in lines 206 to 209 of the Results and touched on again in the Discussion (lines 328 – 332).

- There are over 400 species of which there is sequence information on 12. We are confident about our model, but it could certainly be revisited when sequences are available from substantially more species. 

And then what I really miss in this ms is the sequence data for SP for the Canadian eelpout, would have been beneficial to have that one to see how divergent this SP sequence is compared to the other ones. Any change that the authors could get hold of that sequence information? This information would for sure enlighten the evolutionary path of the SP within this lineage.

- Unfortunately, the only sequence available from this fish is a QAE isoform and this was determined by Edman degradation of a purified protein in 1987 (Schrag JD, Cheng C-HC, Panico M, Morris HR, DeVries AL. Primary and secondary structure of antifreeze peptides from arctic and antarctic zoarcid fishes. Biochim Biophys Acta - Protein Struct Mol Enzymol. 1987;915: 357–370). 

My last comment is about the speculation how the eelpouts have migrated down south. Is there not a possibility that the eelpouts have moved northwards and southwards from different refugia? And diversified in different rounds after settlement in the different regions? Would have loved to see some more speculations here – and then not at least linked more to past paleoclimatic events during the Miocene (cooling ++ as well as different possible refugia during that period). Think this explanation is more likely to have found place.

- We think it is unlikely for two reasons stated in the manuscript: 1) The AFPs are all clearly related and appear to have evolved once. 2) The Antarctic species arose from northern species long after ice was again present at the poles, so the fish had clearly left refugia and populated icy seas well before the lineage that includes the southern species arose. 

In relation to this, they write that the eelpouts might have migrated via the cold depths and at the same time (on the way down) lost their SP due to lack of ice crystals at these depths. Not sure if the “red” color on Figure 4 is describing this in a good way as well as the migratory route (not that illustrative). Would suggest to modify this figure and also take into account the other possibilities re coming for different refugia f. ex.

- The red colour indicates the warmer surface waters near the equator, whereas the blue indicates cold water at all depths in the north and south and at depths throughout the tropics. This is now clearly stated in the legend to Fig. 4. We have added a dashed arrow to illustrate the most likely direction of migration. We have not added the other speculation to the figure to avoid confusing the reader but have detailed this in the text as described above. 

One minor comment:

Would strongly suggest to use gene duplication not amplification.

- In our earlier work we have published several examples of AFP genes going from a presumed single copy to huge tandem arrays of 30 to over 150 gene copies. Although gene duplication was no doubt responsible for the initial expansion of gene copy number, unequal crossing over would be at work for the major expansion of the tandem arrays. We have now added this definition of gene amplification into the manuscript on lines 235 and 236. 

“This suggests that their genes have undergone multiple rounds of gene duplication and unequal crossing over (gene amplification) within the last few million years, after the ocean pout lineage separated from that leading to the notched fin and viviparous lineages.” 

Reviewer #2: Antifreeze proteins are fascinating examples of convergent evolution of function. Four different types of antifreeze proteins have evolved in fish species throughout the globe. This manuscript addresses three questions about the evolution of Type III antifreeze proteins in Zoarcales including whether the proteins arose by direct descent vs parallel evolution, when the antifreeze proteins (AFPs) arose, and the timing of the colonization of the Antarctic by zoarcids.

The finding of the AFP in the Alaskan ronquil is exciting and challenges the current accepted timeline for the evolution of AFPs. I agree with the manuscript that it also provides additional evidence for an earlier cooling of the Arctic Ocean.

The evidence for the evolution of QAE and SP isoforms is contradictory. While the phylogenetic tree places the appearance of the QAE at the Bathymasteridae (Figure 1), the tree of relationships in Figure 3 suggests that the SP isoform is ancestral and the QAE isoform is the derived state, especially with regards to the placement of the OPpan and the AEsasB. To simply state it: Figure 1 shows QAE arose first and SP is derived whereas Figure 3 shows the opposite (SP is ancestral and QAE is derived). The results in Figure 4 are also contradictory with the statements regarding the Antarctic having the highly derived sequence. Many of the branches have little to no support, which suggests the branches should be collapsed and not interpreted as evidence for the findings. Moreover, the authors propose that “likely lost all but one or a few nearly-identical AFP genes during the journey [from the Arctic to Antarctic]” and that “gene losses and duplications have occurred frequently, particularly within the SP group” so it is possible that SP were lost in other lineages and the SP form arose first.

- We argue that there is no evidence to support one isoform type preceding the other and have added some clarifications to address the concern of the reviewer. Our lack of recovery of an SP isoform from ronquil could have occurred due to non-matching primers or the poor quality of the museum specimen used. We have altered figure 1 to include “SP?” at the branchpoint leading to ronquil to indicate that it is still undetermined as to when this isoform arose. We would also argue that figure 3 does not indicate that SP arose first. The SP and QAE groups diverge at a single node. If the two “intermediate” sequences are excluded, the bootstrap value for this node is very well supported. If the QAE sequences arose from an SP sequence, we would expect them to form a subcluster within the SP tree, rather than being separate from it.

The manuscript highlights a recent gene amplification in the Viviparous eelpout, however, there is no discussion of the L. dearborni expansion. There is data available from ~30 L. dearborni AFPs on GenBank through BAC sequencing (Deng et al 2010), which has the genomic context for the expansion.

- The sequences from these BACs were indeed used in our manuscript, the accession numbers were given, and the manuscript was cited. We have added text in the Discussion (line 390) and in Materials and Methods (line 448) to make it clearer that all these BACs were examined, and we have cited the manuscript again there and on line 357. The reason that only seven L. dearborni sequences were used in the phylogenetic tree was that most of these sequences differed at two or fewer residues, so they would not have added anything of value to the phylogenetic analysis. We do state in Materials and Methods that this criterion was used when selecting sequences for further analysis but had not mentioned L. dearborni specifically as it was applied to sequences from all species.

The findings are generally confounded by the duplications and deletions that are presumably occurring very frequently.

- On line 240, we reference the example of type III AFP gene copy variation in different populations of present-day eel pouts (see reference 26).

For some of the sections, the authors provide alternative explanations. The alternative explanations are incredibly helpful in interpreting the results and I appreciate the inclusion.

- We appreciate that comment.

The figure legends and text are confounded and it is hard to determine what is in the figure legend and what is main text.

- We have paid particular attention to remove results and discussion from figure legends and have only placed there information needed to interpret the figure.

Initials of a common name seems to be an unusual way to label sequences in the text and Figure 2 (sequence alignment).

- As requested by Reviewer 1 we have changed the labeling to refer now to common species names.

The tissue of origin is included on some of the isoforms, does the tissue of origin matter? The liver is the most common place for transcription of AFPs but there seem to be tissue-specific transcripts. Are those relevant for the organismal performance? There are several similar smaller results in the manuscript where it is unclear the relevance to the larger findings.

- It is difficult to say if the tissue origin will be important at some point. This information has been removed from the labels and some discussion of tissue origin has been removed from Results. However, this information can still be accessed from Table S3.

None of the supplementary tables are available.

- Our apologies for this oversight. They are now provided.

Figure S4 is not referred to in the main text and is not available in the supplementary materials.

- We reference this figure on lines 303-304 of the manuscript. We do apologize that this figure was somehow not included in the draft pdf that the reviewers received.

Is there evidence for loss of the SP isoform?

- Only as much as the SP isoforms are not found in those species in Antarctica but they are present in the northern species.

Figure 4C is not informative.

- The description for this part of the figure was lacking in the legend. We have rectified this. It represents a cross-section of the oceans from north to south through which the eel pouts migrated in deep, cold water. Water temperature is indicated by a red (warm) – blue (cold) gradient. The likely direction of migration is indicated by a dotted arrow.

There are some references that are missing (for example, on line 111).

- We have checked the source of every nucleotide and protein sequence from ocean pout in the NCBI databases and the three references we cited account for all of them. 

Line 269 has some specific PLOS guidelines that should be removed.

- Done

We trust that these changes will meet with your approval.

Sincerely,

Peter L. Davies, PhD, FRSC

Canada Research Chair in Protein Engineering

Laurie A. Graham, PhD

---

## [Decision Letter · Decision Letter 1]

17 Jun 2020

PONE-D-19-34926R1

Antifreeze protein dispersion in eelpouts and related fishes reveals migration and climate alteration within the last 20 Ma

PLOS ONE

Dear Dr. Davies,

Thank you for submitting your manuscript to PLOS ONE. After careful consideration, we feel that it has merit but does not fully meet PLOS ONE’s publication criteria as it currently stands. Therefore, we invite you to submit a revised version of the manuscript that comprehensively addresses the points raised during the review process.

We look forward to receiving your revised manuscript.

Kind regards,

Michael Schubert

Academic Editor

PLOS ONE

Reviewers' comments:

Reviewer's Responses to Questions

**Comments to the Author**

1. If the authors have adequately addressed your comments raised in a previous round of review and you feel that this manuscript is now acceptable for publication, you may indicate that here to bypass the “Comments to the Author” section, enter your conflict of interest statement in the “Confidential to Editor” section, and submit your "Accept" recommendation.

Reviewer #1: (No Response)

Reviewer #2: All comments have been addressed

2. Is the manuscript technically sound, and do the data support the conclusions?

Reviewer #1: Yes

Reviewer #2: Yes

3. Has the statistical analysis been performed appropriately and rigorously? 

Reviewer #1: N/A

Reviewer #2: No

4. Have the authors made all data underlying the findings in their manuscript fully available?

Reviewer #1: Yes

Reviewer #2: Yes

5. Is the manuscript presented in an intelligible fashion and written in standard English?

Reviewer #1: Yes

Reviewer #2: Yes

6. Review Comments to the Author

Reviewer #1: Dear Dr. Hobbs et al.,

I have now read through the revised version of the ms PONE-D-19-34926 in detail, and appreciate the fact that the authors have taken many of my comments into account, but I still have some issues that I think should be dealt with before acceptance for publication.

First, I think the authors did a great job on the introduction and results section, it reads so much better now. Also, happy to see that they performed the suggested additional analyses testing for positive and/or negative selection.

Re the figures I do see that they have tried to change accordingly to my suggestions – but still think they could be improved a bit further:

It is the shortening of some of the common names + underline + underscore that bothers me the most.

I strongly advise the authors to come up with a better naming system – where f. ex. the number for the different sequences in the same cluster is the same (not following a number given by you and linked to the accession number): like “common name” followed by QEA1, “common name” followed by QEA2 and so on – which could be easily used in the text as well – and much more meaning full. The linking of the sequence number to the accession number should still be done – but after you have done the numbering of the clusters – hope you get what I mean here.

I see that this will take up some space but still better than as is with the shortening of the names + the underscore + the number (linked to the accession numbers).

Would be so much better if the sequences are proper qualified into different clades/clusters.

Think this would be less confusing, like the short names + the number linked to the sequences which are used in the text (for example ronquil-1 and Ant_eelpout_4 +++).

Then I would also strongly suggest to use asterisk * or ** instead of underline (think this will look better).

Then to the inference and discussion of the results:

When I now re-read the discussion – I do see that the discussion has become more coherent – but think it still could benefit some re-writings. F.ex. it takes a few paragraphs before they start the discussion of their results for real. I would have started the different paragraphs with a highlight of the results – before going into the discussion – then the reader gets the message/new findings up front.

For dN/dS the results – discussed in line 324-337 – I think the authors could go a bit further in their interpretation here regarding these findings – since they do find differences in positive selection for the QAE vs SP sequences. This is not highlighted in the discussion – which I think it should: the higher degree of positive selection found for the QAE sequences could be linked to the higher variation observed as well as in regards to the clustering patterns. It is stated somewhat in the discussion, but not in relation to the difference in dN/dS ratio observed between the QAE and SP sequences. I truly think this could be linked to the weaker association to families by the QAE sequences – as compared to the SP – which indeed show a clearer separation by family (i.e. which then could be linked to the lower degree of similar positive selection). See my comments below (too (also related to this issue).

Line 329: Can you please elaborate on your results and function of the signatures of selection that you find on the surfaces?

Do understand that they are not involved in the binding to the pyramidal ice plane which are found to be conserved. Are they found on the prism IBS? What is the function of the prism IBS? If not, where do you find most changes – and any idea what those changes could have of effect on the function?

Line 331-333:

I find this sentence a bit awkward and suggest to re-write:

“This together with the fact that the type III AFT is only found within this linage, and not a result of gene conversion and/or lateral gene transfer, indicates that …..”

In the first review I stated the following:

“I would say that the higher divergence between species for the QAE sequences indicate strongly that they origin from the same ancestor QAE sequence, that have then evolved in the different lineages in different paces/slightly different directions. But for SP the results could indicate that the SP sequences have evolved over and over again independently for the different families (from the SAS sequence specific for the different species and/or the common ancestor for the specific families). Think this cannot be ruled out – and maybe also plausible – since the SPs is only functional in combination with the QAE type (and could be looked upon as added value in addition to the QAE under certain circumstances I guess).”

You say that you have added this – cannot see that this has been included (in more depth than already mentioned in line 226-228 (was already included in the ms the first round).

Further I stated: “And f. ex not sure if I truly agree on the statement re the QAE sequences on line 255-256: “This suggests that the common ancestor of these families may have possessed a larger number of QAE sequences than SP sequences.””

How I see this now (when the dN/dS analyses have been performed): Is that both groups (QAE and SP) have most likely undergone gene duplications – also the QAE (as you indirectly state by saying that they have a large number from the very beginning). It is most likely the timing and fate of the gene duplication for the different groups that is somewhat different, with more and stronger selection for the QAEs compared to the SPs, which again could imply that the QAE gene duplications are more likely to be maintained and also result in genes with more similar function over different families. This goes for SP too but most likely at a slower pace. Here we have could have gene duplications that are more easily lost and that selection for similar function is maybe not that evident? In fact, the lower degree of positive selection -> indicate a lower degree of gene duplications (that are maintained) not higher as the authors state. The authors should look into those statements.

For instance, you write in line 371-372: “This suggests that gene losses and duplications have occurred frequently, particularly within the SP group, within the last 10 Ma.”

Can you really say this? Think we for sure agree that this is a result of gene duplication events -> but can we really infer from this that they happened more frequently?

Additionally, is it so that the duplications/precursors for the QAE arose in the ancestor of this lineage, while for the SP sequences the duplication event could have occurred later (in separate rounds, as I stated in my previous comment to the authors). To infer this, it would actually been nice to see the different variants found of both QAE and SP mapped onto the species tree, and also be a nice add on as one of the main figures of the paper. And if made, inference of the timing can be made.

Think the authors should revisit the ms with this in mind and add comments re the different possibilities both in the results section as well as the discussion.

Then I state in my first review:

“These speculations, as well as about more gene losses and duplications being responsible for the clustering of the SP sequences can only be inferred if they have had whole genome sequencing data. Same goes for the plausible loss of the SP in the Antarctica species. Full genome data set is needed to confirm a loss so should be careful here I think.”

This was a statement to the authors that precaution should be made – since they do not have the full overview of the gene variant present or not – and that this should be stated in the paper (i.e. that full genome data-sets are needed to look into this in more detail).

And PS: fully aware of that these genes can be hard assemble – but they will be part of the raw reads and/or unassembled contigs (so most likely genome data will aid here too even if not fully put together).

Furthermore, I stated in my pervious review:

“Additionally, in this paper only two Antarctic species are investigated (it is listed that this linage contain about 100 species (so I guess some of them could have the SP type (i.e. not yet confirmed but then either retained if lost in some of them and/or evolved from the SAS sequence and/or the QAE sequence in the common ancestor of this lineage)).”

This needs to be addressed I think (still not done I see) – that you have only looked into two species – and that the loss observed could potentially not be the case for all of these species.

Moreover, I also stated:

“And then what I really miss in this ms is the sequence data for SP for the Canadian eelpout, would have been beneficial to have that one to see how divergent this SP sequence is compared to the other ones. Any change that the authors could get hold of that sequence information? This information would for sure enlighten the evolutionary path of the SP within this lineage.”

I see that this in not obtained – but they do state that a QAE isoform is identified – can I ask why is this then not included in your analyses? This would also have been a nice add on to the paper for sure! And also, I would have appreciated if the authors could have mentioned made the authors aware of the fact that SP is not obtained in the Canadian eelpout. Could it be that it is also lacking from this species -> lost in the common ancestor? Or do you know that it is present? If yes, how do you know this? In other words – some more elaboration around this – the findings vs. interpretations and limitations in their dataset is still needed.

And I do see that the authors did not agree with me re my hypothesis re a common refugia. How was the climate 5 mill years ago? All fish settled in the north they think?

Would still have loved to see some more speculations here – and then not at least linked more to past paleoclimatic events during the Miocene (cooling ++ as well as different possible refugia during that period). Any reason for why some should start migrating down south?

My last and final comment, is that I miss a concluding remark, as is the discussion is ending quite abruptly.

Some minor comments:

In line 350 you state:

“The two Antarctic eelpouts diverged from the Canadian eelpout more recently than 10 Ma (Fig 1).”

This is not correct I think: the Antarctic eelpouts and the Canadian eelpots have a common ancestor -> they did clearly not diverge from the Canadian eelpouts. Hop ethe authors agree – and re-write.

In line 357 you state:

“The population that migrated south, ....”

This is pure speculation ….. how the migration occurred we do not know …. if it was one or more populations …... could it have been several events? And not at least, the specimens that migrated could have diverged along the gradient down south. Why I say this is that you so far only have two of the Antarctic species included in this study.

The only thig that needs to be added is that “in the species investigated we find …..” so that the reader is aware of this.

In line 381 you state:

“The two Antarctic species did not retain SP isoforms as mentioned above, .....”

Do not get as mention above – in the result section? Not stated anywhere in the discussion as I can see…. Think you should re-write.

Reviewer #2: The revised manuscript is much improved from the original submission.

There are a few outstanding / new issues with the manuscript but I think they are all straightforward to address.

The section on positive selection is incorrect. The added dn/ds analysis is incorrect / incomplete. There are several issues that I outline here: SNAP does not test for selection, so in the current implementation dn/ds > 1 cannot be distinguished from dn/ds = 1. To test for selection, the authors will need to use hyphy or codeml or another similar program that actually uses different codon models to test for selection. The cumulative dn and ds rates (Fig 10A) is not informative about positive selection. It is notable that the signal peptide does not have nonsynonymous or synonymous substitutions, but this has been observed before. The pairwise comparisons with ten or more positions likely leads to an overestimate of dn/ds.

Results line 201+ The inclusion of only the subset of sequences that are at least three amino acids different makes some sense in the figure but the subsequent discussion in the results seems to be about all sequences so there is an inherent challenge in reading the paragraph in the results.

The relevance of the comment about no tandemers in P. brachycephalum in the introduction is unclear.

The end of the first paragraph of the results seems to contradict the introduction about what is known about the shanny and gunnel.

Line 118 three Zoarcales lineages are referred to but it’s not entirely clear which three are relevant here.

Figure 4 was an issue for both reviewers and I still do not feel it is particularly clear and it does not represent any alternative possibilities.

Figure 1, Fig S2 (and possibly others), the underlining of either genus or genus species is not clear to me. The statement in the figure legend does not clarify.

Line 138, only needs to reference Fig 2 for the ronquil-1 sequence.

Line 142, unclear why the percent similarity is a range when it is a single pairwise comparison.

Lines 150-152, the meaning of the sentence is not entirely clear.

Line 199, include the tree in the supplement.

Line 329, should be substitution not “mutation” the dn/ds method analyzes substitutions not mutations.

Line 389, issue with citation format

The figure legends are helpful but now quite extensive and could be shortened for clarity and brevity.

7. PLOS authors have the option to publish the peer review history of their article (what does this mean?). If published, this will include your full peer review and any attached files.

Reviewer #1: No

Reviewer #2: No

---

## [Author Response · Author response to Decision Letter 1]

17 Nov 2020

Dr. Michael Schubert

Academic Editor

PLOS ONE

Dear Dr. Schubert,

Once again, I would like to thank you for a second opportunity to revise our manuscript PONE-D-19-34926R1 and for giving us an adequate time frame in which to do this. We note that Reviewer #2 was satisfied with our previous responses, but we have still addressed their remaining queries and corrections. We have accommodated most of Reviewer #1’s criticisms but beg to differ on a few issues. All the changes we made are documented in point-by-point form in blue after the Reviewers’ queries.

We trust these changes will allow this study to be published.

With kind regards,

Peter L. Davies

Reviewers' comments:

Reviewer's Responses to Questions

Comments to the Author

1. If the authors have adequately addressed your comments raised in a previous round of review and you feel that this manuscript is now acceptable for publication, you may indicate that here to bypass the “Comments to the Author” section, enter your conflict of interest statement in the “Confidential to Editor” section, and submit your "Accept" recommendation.

Reviewer #1: (No Response)

Reviewer #2: All comments have been addressed

2. Is the manuscript technically sound, and do the data support the conclusions?

Reviewer #1: Yes

Reviewer #2: Yes

3. Has the statistical analysis been performed appropriately and rigorously?

Reviewer #1: N/A

Reviewer #2: No 

4. Have the authors made all data underlying the findings in their manuscript fully available?

Reviewer #1: Yes

Reviewer #2: Yes 

5. Is the manuscript presented in an intelligible fashion and written in standard English?

Reviewer #1: Yes

Reviewer #2: Yes 

6. Review Comments to the Author

Reviewer #1: Dear Dr. Hobbs et al.,

I have now read through the revised version of the ms PONE-D-19-34926 in detail, and appreciate the fact that the authors have taken many of my comments into account, but I still have some issues that I think should be dealt with before acceptance for publication.

First, I think the authors did a great job on the introduction and results section, it reads so much better now. Also, happy to see that they performed the suggested additional analyses testing for positive and/or negative selection.

Re the figures I do see that they have tried to change accordingly to my suggestions – but still think they could be improved a bit further:

It is the shortening of some of the common names + underline + underscore that bothers me the most.

I strongly advise the authors to come up with a better naming system – where f. ex. the number for the different sequences in the same cluster is the same (not following a number given by you and linked to the accession number): like “common name” followed by QEA1, “common name” followed by QEA2 and so on – which could be easily used in the text as well – and much more meaning full. The linking of the sequence number to the accession number should still be done – but after you have done the numbering of the clusters – hope you get what I mean here.

I see that this will take up some space but still better than as is with the shortening of the names + the underscore + the number (linked to the accession numbers).

Would be so much better if the sequences are proper qualified into different clades/clusters.

Think this would be less confusing, like the short names + the number linked to the sequences which are used in the text (for example ronquil-1 and Ant_eelpout_4 +++).

Then I would also strongly suggest to use asterisk * or ** instead of underline (think this will look better).

We have followed the reviewer’s suggestions for improving the nomenclature of species and sequences. Thus in Figure 1, underlines have been replaced by single and double asterisks to denote the source of the taxonomic determination. Dashes and underscores have been removed from Figure S1 and underlines from all names in all Figures. As requested, abbreviated common names have been replaced by full common names in Figures 2, 3, 4, S3, S4, S5, S7, S9 and Tables S3 and S4. The letter Q or S has been added to each name to differentiate QAE and SP isoforms respectively as this keeps the names slightly shorter. For each species, they have been numbered consecutively within each group as they appear in the tree. As the two main groups within the QAE group cluster by species groups (northern or southern), they were not further differentiated. The text of the main manuscript has been modified to match. For example, P. brachy-1 is now P. brachycephalum-Q1 and viv-eelpout-12 is now viviparous eelpout-S7.

Then to the inference and discussion of the results:

When I now re-read the discussion – I do see that the discussion has become more coherent – but think it still could benefit some re-writings. F.ex. it takes a few paragraphs before they start the discussion of their results for real. I would have started the different paragraphs with a highlight of the results – before going into the discussion – then the reader gets the message/new findings up front.

A paragraph introducing the focus of the paper has been added to the beginning of the Discussion (lines 332-338). 

For dN/dS the results – discussed in line 324-337 – I think the authors could go a bit further in their interpretation here regarding these findings – since they do find differences in positive selection for the QAE vs SP sequences. This is not highlighted in the discussion – which I think it should: the higher degree of positive selection found for the QAE sequences could be linked to the higher variation observed as well as in regards to the clustering patterns. It is stated somewhat in the discussion, but not in relation to the difference in dN/dS ratio observed between the QAE and SP sequences. I truly think this could be linked to the weaker association to families by the QAE sequences – as compared to the SP – which indeed show a clearer separation by family (i.e. which then could be linked to the lower degree of similar positive selection). See my comments below (too (also related to this issue).

The dN/dS analysis was added at the reviewer’s request. However, we realize this type of analysis is beyond our expertise. With Reviewer 2 finding fault with our dN/dS analysis, we have withdrawn this section and instead have expanded our analysis of the effects of mutations on the protein, as per the reviewer’s request (see paragraph below). We suggest either reviewer could consider performing a dN/dS analysis of the sequences herein and publish their analysis. We would be happy to provide any additional sequences needed.

Line 329: Can you please elaborate on your results and function of the signatures of selection that you find on the surfaces?

Do understand that they are not involved in the binding to the pyramidal ice plane which are found to be conserved. Are they found on the prism IBS? What is the function of the prism IBS? If not, where do you find most changes – and any idea what those changes could have of effect on the function?

We have expanded our interpretation of the mutations that are found on the surfaces of the AFPs by adding a section addressing the hydrophobicity of the AFPs relative to the progenitor (starting at line 303) and we have somewhat modified the section starting on line 282. Lines 465-480 of the Discussion have been modified as well. We have also expanded figure S10 to included SAS and the shift in hydrophobicity. We also discuss the difficulty with assessing mutations and their effect on activity starting on line 489.

Line 331-333:

I find this sentence a bit awkward and suggest to re-write:

“This together with the fact that the type III AFT is only found within this linage, and not a result of gene conversion and/or lateral gene transfer, indicates that …..”

We could not find this statement in our most recent submission.

In the first review I stated the following:

“I would say that the higher divergence between species for the QAE sequences indicate strongly that they origin from the same ancestor QAE sequence, that have then evolved in the different lineages in different paces/slightly different directions. But for SP the results could indicate that the SP sequences have evolved over and over again independently for the different families (from the SAS sequence specific for the different species and/or the common ancestor for the specific families). Think this cannot be ruled out – and maybe also plausible – since the SPs is only functional in combination with the QAE type (and could be looked upon as added value in addition to the QAE under certain circumstances I guess).”

We agree that it is difficult to ascertain the evolutionary forces that are generating the divergence within the type III gene family. Whenever genes are present in a large gene family, it is challenging to try to tease apart the role of gene duplication/amplification and gene loss, as well as genetic drift and founder effects, from selection. However, from Figure 2 it is apparent that the SP sequences from all the species share similarities that are unique to this group and that are not found within the QAE sequences. Additionally, they cluster in Figure 3. Thus we see nothing to suggest SP sequences have evolved over and over again.

You say that you have added this – cannot see that this has been included (in more depth than already mentioned in line 226-228 (was already included in the ms the first round).

The changes we made were in the Discussion. These issues are discussed in the paragraphs starting on line 370, 403, 422 and 444, which have been further modified. We have also added an analysis including wolf eel SAS to the Results, within the paragraph starting on line 213.

Further I stated: “And f. ex not sure if I truly agree on the statement re the QAE sequences on line 255-256: “This suggests that the common ancestor of these families may have possessed a larger number of QAE sequences than SP sequences.””

This is our conjecture based on the greater diversity of the QAE isoforms, particularly in the ocean pout. The statement is doubly qualified with the use of “suggests” and “may have”. We cannot be certain what happened millions of years ago and is still going on, but have put forward what we think is the most plausible scenario. That there are other possible scenarios is not grounds to obstruct publication of this work. Again, these other ideas could be published in a follow-up paper or could be revisited once the technology improves to the point that we can obtain complete assemblies of AFP loci from multiple fish. 

Please note: PLOS now offers accepted authors the opportunity to publish the peer review history of their manuscript alongside the final article. The peer review history package includes the complete editorial decision letter for each revision, with reviews, and our responses to reviewer comments. We would like to pick this option so that your alternative suggestions can be aired.

How I see this now (when the dN/dS analyses have been performed): Is that both groups (QAE and SP) have most likely undergone gene duplications – also the QAE (as you indirectly state by saying that they have a large number from the very beginning). It is most likely the timing and fate of the gene duplication for the different groups that is somewhat different, with more and stronger selection for the QAEs compared to the SPs, which again could imply that the QAE gene duplications are more likely to be maintained and also result in genes with more similar function over different families. This goes for SP too but most likely at a slower pace. Here we have could have gene duplications that are more easily lost and that selection for similar function is maybe not that evident? In fact, the lower degree of positive selection -> indicate a lower degree of gene duplications (that are maintained) not higher as the authors state. The authors should look into those statements.

For instance, you write in line 371-372: “This suggests that gene losses and duplications have occurred frequently, particularly within the SP group, within the last 10 Ma.”

Can you really say this? Think we for sure agree that this is a result of gene duplication events -> but can we really infer from this that they happened more frequently?

Additionally, is it so that the duplications/precursors for the QAE arose in the ancestor of this lineage, while for the SP sequences the duplication event could have occurred later (in separate rounds, as I stated in my previous comment to the authors). To infer this, it would actually been nice to see the different variants found of both QAE and SP mapped onto the species tree, and also be a nice add on as one of the main figures of the paper. And if made, inference of the timing can be made.

Think the authors should revisit the ms with this in mind and add comments re the different possibilities both in the results section as well as the discussion.

The results within even a single species (notably Macrozoarces americanus for example) indicate that gene amplifications are ongoing and that there can be a large number of different isoforms. Also, due to the difficulties of assembling and characterizing multigene families, the isoform assemblages of most of the species examined herein are incomplete. Therefore, it would be impossible to ascertain the fate of individual gene copies within the species examined in this study. However, there are indications that gene losses and/or gene duplications/amplifications of either QAE or SP isoforms have occurred in specific lineages. To this end, we have mapped these events onto the species tree by adding SP↑ to two locations in Figure 1 to indicate branches were amplification of the SP complement has occurred. The possible locations where the SP and QAE arose or were lost were already present in this figure. 

Then I state in my first review:

“These speculations, as well as about more gene losses and duplications being responsible for the clustering of the SP sequences can only be inferred if they have had whole genome sequencing data. Same goes for the plausible loss of the SP in the Antarctica species. Full genome data set is needed to confirm a loss so should be careful here I think.”

This was a statement to the authors that precaution should be made – since they do not have the full overview of the gene variant present or not – and that this should be stated in the paper (i.e. that full genome data-sets are needed to look into this in more detail).

And PS: fully aware of that these genes can be hard assemble – but they will be part of the raw reads and/or unassembled contigs (so most likely genome data will aid here too even if not fully put together).

Papers have been published outlining the difficulties with assembling multi-gene families, especially using short-read techniques where genome assemblies fail to generate AFP loci (see Zhuang X, Yang C, Fevolden SE, Cheng CH. Protein genes in repetitive sequence-antifreeze glycoproteins in Atlantic cod genome. BMC Genomics. 2012, 13:293, PMID: 22747999. For this reason, Zhang et al. opted to isolate BAC clones from one Antarctic species (Zhang, J., Deng, C., Wang, J. et al. Identification of a two-domain antifreeze protein gene in Antarctic eelpout Lycodichthys dearborni. 2009 Polar Biol 32:35). All of the type III sequences are similar enough to be detected using probes to either one, so the lack of an SP isoform in any of the BAC sequences would suggest that it is not present. 

Additionally, the second Antarctic species was analysed at both the protein and transcriptome level, so if any SP genes are present, they do not appear to be expressed. Furthermore, the recent release of the Anarrhichthys ocellatus (wolf-eel) genome sequence demonstrates the problem in spades. The four putative AFP genes that were assembled are on scaffolds ~2 kb length. However, both SAS genes were assembled and lie next to each other on a 5.8 Mb scaffold. These sequences have been added to both Figure 2 and Figure 3, as well as to supplementary figures, and they have been used to addresses concerns raised about the possibility that these sequences could have arisen anew from SAS. 

Furthermore, I stated in my pervious review:

“Additionally, in this paper only two Antarctic species are investigated (it is listed that this linage contain about 100 species (so I guess some of them could have the SP type (i.e. not yet confirmed but then either retained if lost in some of them and/or evolved from the SAS sequence and/or the QAE sequence in the common ancestor of this lineage)).”

This needs to be addressed I think (still not done I see) – that you have only looked into two species – and that the loss observed could potentially not be the case for all of these species.

For our analysis we have combed the database and have included as many sequences as we could find from other studies. This is admittedly a small sampling of the over 300 Zoarcid species, including those from northern waters, but ours is the first study to combine a dataset of this size. In order to achieve this, we would have to go to the Antarctic waters and collect dozens of species and sequence and assemble their genomes using a long-read platform such as PacBio, in order to bolster some reasonable speculation. While this would be ideal, it is not feasible. 

Furthermore, the large number of sequences obtained by transcriptome sequencing of viviparous eelpout surely allows us to draw some inferences about the diversity of SP sequences present in this species as over 2000 reads were obtained. Liver is the dominant site of AFP production in all species examined, such as eelpouts and wolffish (line 144) and shanny and gunnel (Fig S6). SP sequences have also been obtained from the liver of ocean pout (see table S3 for example) yet these cluster with those obtained from other tissues and separate to those recovered from viviparous eelpout. If lineage-specific gene amplifications had not occurred, this would be unexpected.

Moreover, I also stated:

“And then what I really miss in this ms is the sequence data for SP for the Canadian eelpout, would have been beneficial to have that one to see how divergent this SP sequence is compared to the other ones. Any change that the authors could get hold of that sequence information? This information would for sure enlighten the evolutionary path of the SP within this lineage.”

This sequence was obtained by Edman degradation of a protein and was reported by Schrag et al. in 1987. We do not have access to this particular fish at this time. As a nucleotide sequence was not available, this sequence was not used in the generation of the main trees within this manuscript, but it was always shown in Fig S3. We have now added a supplementary tree (Fig S7) that shows that this sequence clusters with the sequences from the closely-related Antarctic fish. 

We feel that the inclusion of this information is merited as this species is more closely related to the Antarctic species than any of the other species for which we have sequence data. Unfortunately, while we know that this species has more than one AFP isoform, given that the manuscript by Schrag et al. in 1987 shows three active protein peaks, we do not know anything substantive about them as only one was subjected to further analysis. We have included this information in the Discussion. While it would be nice to have additional sequence information from this species, we do not have access to this fish at this time and we do not know if it has SP isoforms.

I see that this in not obtained – but they do state that a QAE isoform is identified – can I ask why is this then not included in your analyses? This would also have been a nice add on to the paper for sure! And also, I would have appreciated if the authors could have mentioned made the authors aware of the fact that SP is not obtained in the Canadian eelpout. Could it be that it is also lacking from this species -> lost in the common ancestor? Or do you know that it is present? If yes, how do you know this? In other words – some more elaboration around this – the findings vs. interpretations and limitations in their dataset is still needed.

See above. 

And I do see that the authors did not agree with me re my hypothesis re a common refugia. How was the climate 5 mill years ago? All fish settled in the north they think?

Would still have loved to see some more speculations here – and then not at least linked more to past paleoclimatic events during the Miocene (cooling ++ as well as different possible refugia during that period). Any reason for why some should start migrating down south?

We have added more discussion of this topic. The Zoarcids were undergoing rapid speciation during the last ice age, perhaps because their AFP genes allowed them to thrive in a glacial epoch ocean? Given the extent of the ice sheets, their range was undoubtedly shifted southward, but it would be very difficult to determine the historical localization of so many species over the entire glacial period. However, we have expanded our discussion of previous work that addresses the distribution of early Zoarcids. That being said, during the peak of the last ice age, the tropical surface ocean waters were only a few degrees cooler than they are today, so they could still have acted as a barrier to the migration of fishes adapted to cold water. We did neglect to explicitly state this and have added a reference to address this. See lines 356-369 in the Discussion. 

With regards to migration, our hypothesis is that the deep ocean environment around features such as hydrothermal vents is not dependent on latitude. Zoarcids are the dominant fish at such features. Therefore, they may have simply populated the entire length of the divergent plate boundaries that span the ocean floors. Unfortunately, this is merely a correlation and we cannot directly prove causation. We have added material starting around line 384 of the Discussion to bolster this hypothesis. 

My last and final comment, is that I miss a concluding remark, as is the discussion is ending quite abruptly.

The final paragraph of the Discussion has been altered to provide the missing concluding remarks.

Some minor comments:

In line 350 you state:

“The two Antarctic eelpouts diverged from the Canadian eelpout more recently than 10 Ma (Fig 1).”

This is not correct I think: the Antarctic eelpouts and the Canadian eelpots have a common ancestor -> they did clearly not diverge from the Canadian eelpouts. Hop ethe authors agree – and re-write.

This was sloppy wording on our part. We should have said: “The two Antarctic eelpouts diverged from the Canadian eelpout lineage more recently than 10 Ma (Fig 1).”

The section containing this statement has been rewritten as follows (starting on line 413).

…. Canadian eelpout, which is the closest relative to the Antarctic species in our study, with these two lineages having diverged less than 10 Ma (Fig 1).

In line 357 you state:

“The population that migrated south, ....”

This is pure speculation ….. how the migration occurred we do not know …. if it was one or more populations …... could it have been several events? And not at least, the specimens that migrated could have diverged along the gradient down south. Why I say this is that you so far only have two of the Antarctic species included in this study.

The only thig that needs to be added is that “in the species investigated we find …..” so that the reader is aware of this.

In line 431 of the Discussion we have noted that other species may have migrated south and may have retained SP isoforms and that there may have been multiple migrations to southern waters.

In line 381 you state:

“The two Antarctic species did not retain SP isoforms as mentioned above, .....”

Do not get as mention above – in the result section? Not stated anywhere in the discussion as I can see…. Think you should re-write.

The Discussion concerning the Antarctic species has been largely rewritten, so this statement (now on line 459), is referring to the paragraph beginning on line 422. 

Reviewer #2: The revised manuscript is much improved from the original submission.

There are a few outstanding / new issues with the manuscript but I think they are all straightforward to address.

The section on positive selection is incorrect. The added dn/ds analysis is incorrect / incomplete. There are several issues that I outline here: SNAP does not test for selection, so in the current implementation dn/ds > 1 cannot be distinguished from dn/ds = 1. To test for selection, the authors will need to use hyphy or codeml or another similar program that actually uses different codon models to test for selection. The cumulative dn and ds rates (Fig 10A) is not informative about positive selection. It is notable that the signal peptide does not have nonsynonymous or synonymous substitutions, but this has been observed before. The pairwise comparisons with ten or more positions likely leads to an overestimate of dn/ds.

As indicated in our response to reviewer 1 we are out of our expertise here and have decided to leave this analysis for others to do in follow-up studies.

Results line 201+ The inclusion of only the subset of sequences that are at least three amino acids different makes some sense in the figure but the subsequent discussion in the results seems to be about all sequences so there is an inherent challenge in reading the paragraph in the results.

If we were to include all of the know sequences from each species, the phylogenetic trees would be very crowded. The exclusion of identical or near identical sequences does not change the gist of the paper. For example, the percentage ranges as given starting on line 215 go to 100% (between 80-100%) or are over 73% within the QAE or SP groups, respectively, whether or not the excluded sequences are included. Nor does the exclusion change the identity ranges between the different groups as the selection criterion meant that the more divergent sequences were retained in the analysis. Furthermore, within each section, the number of sequences recovered and the number retained for the phylogenetic analysis is clearly stated, as for example, in lines 186-192 as shown below.

“A total of 19 unique sequences (12 SP, 7 QAE) were unambiguously assembled from these reads. One exactly matched a previously known protein sequence (AGM97733), while two others differed from ABN42204 and ABN42205 at two and three residues, respectively. Once sequences with two or fewer a.a. differences were excluded, 5 QAE and 8 SP sequences remained, designated viviparous eelpout-Q1 to -Q5 and -S3 to -S10 (Fig S3).”

The relevance of the comment about no tandemers in P. brachycephalum in the introduction is unclear.

To make the relevance clearer, we changed that section from …..

The AFP complement of the Antarctic eelpout has been studied through a combination of protein, cDNA and genomic DNA sequencing (yielding over 20 gene sequences) and it produces both monomers and tandemers consisting of two or more linked AFP domains [23,36–40]. In contrast, there is no evidence of tandemers in P. brachycephalum [39,41]. 

To (starting on line 86) ….

The AFP complement of the Antarctic eelpout has been studied through a combination of protein, cDNA and genomic DNA sequencing (yielding over 20 gene sequences). This species produces both monomers and tandemers consisting of two or more linked AFP domains [23,36–40], whereas the northern species studied produce only monomers. This is a further illustration of how changeable these AFP genes are. Interestingly, there is no evidence of tandemers in the second Antarctic species examined, P. brachycephalum [39,41].

The end of the first paragraph of the results seems to contradict the introduction about what is known about the shanny and gunnel.

This has been changed from ….

iii) the three species newly-examined in this study (Alaskan ronquil, radiated shanny and rock gunnel) diverged prior to the two families (Anarhichadidae and Zoarcidae) from which AFP sequences were previously known.

To (starting on line 114) ….

iii) the three species newly-examined in this study are found in two lineages that diverged ~18 Ma (Alaskan ronquil) and ~15 Ma (rock gunnel and radiated shanny). The two families from which AFP sequences were previously known, Anarhichadidae and Zoarcidae, only diverged from each other ~10 Ma.

Line 118 three Zoarcales lineages are referred to but it’s not entirely clear which three are relevant here.

The names of the three species were added in brackets and the statement was further clarified.

Figure 4 was an issue for both reviewers and I still do not feel it is particularly clear and it does not represent any alternative possibilities.

This thorough review process has forced us to think extensively about our hypothesis. If we thought there was a plausible alternative we would propose it. We previously added to the figure legend to make the story clearer. We have also added material here to the Discussion, starting at line 494, which describes our hypothesis in more detail. We have also added two mentions of this figure to the Discussion at relevant points to allow readers to further understand what is shown. Our hypothesis is really quite simple. A Zoarcid, which developed AFP genes in the North, migrated through the cold deep oceans to Antarctica. Over the time this journey took, AFP genes were lost because they were not under selective pressure. On arrival in ice-laden Antarctic waters, selection for freeze resistance happened again and AFP genes (surviving QAE) were amplified. We have shown this figure and legend to numerous naïve scientists and asked them to explain the figure. They have all given a consistent version along the lines indicated above. We are at a loss to see how it could be made any clearer in the absence of any specific suggestions from the reviewer. 

Figure 1, Fig S2 (and possibly others), the underlining of either genus or genus species is not clear to me. The statement in the figure legend does not clarify.

Underlining denoted whether the matching genus or species was sampled in these studies.

This has been changed to ….

A double asterisk denotes that the fish from these studies was the same as the species in this study, whereas a single asterisk denotes that the fish was a different species within the same genus.

Line 138, only needs to reference Fig 2 for the ronquil-1 sequence.

Done (now line 147).

Line 142, unclear why the percent similarity is a range when it is a single pairwise comparison.

We have made this clearer by changing the passage from this …..

“This sequence is most similar to an Atlantic wolffish QAE isoform (Atl-wolffish-2) with identities between the protein, coding and intronic sequences of 94 to 95%. The 3' UTR is up to 97% identical to other type III sequences.”

To this ….. now on line 149-152.

“This sequence is most similar to Atlantic wolffish-Q2. The identities between the protein sequences, as well as between the coding sequences and the single intron, were between 94 and 95%. The 3' UTR is up to 97% identical to other type III sequences.”

Lines 150-152, the meaning of the sentence is not entirely clear.

This was changed from ….

“Type III AFP sequences from the Atlantic ocean pout (Z. americanus), the species in which this AFP was first discovered, have previously come from protein, cDNA and genomic sequences [25,26,34].”

To this ….. now on lines 159-161.

“Type III AFP sequences from the Atlantic ocean pout (Z. americanus), the species in which this AFP was first discovered, were previously obtained by Edman degradation of proteins or from cDNA and genomic sequences [25,26,34].”

It is important to establish this for the statement on line 171:

Three others (ocean pout-Q5, ocean pout-S2 and ocean pout-S4) match isoforms known only from protein sequencing (HPLC12, HPLC7 and HPLC1, respectively) [26].

Line 199, include the tree in the supplement.

This tree has been added (Fig S7) and in addition, the two AFPs known only from protein sequencing have been added in and their relationship to the other sequences has been described in more detail.

Line 329, should be substitution not “mutation” the dn/ds method analyzes substitutions not mutations.

This sentence has been removed.

Line 389, issue with citation format

Before additional references were added, this citation was to reference 27 and 72, which may have looked to be an inadvertent reversal. Nevertheless, these were the intended references.

The figure legends are helpful but now quite extensive and could be shortened for clarity and brevity.

Accession numbers for the AFPs have been removed from figure legends and placed in table S5. Numerous changes have been made to figure legends to make them more concise. 

For example, in the Figure 2 legend,

Sequences are named using the common name for each species (or a contraction thereof, see Fig S3), except P. brachycephalum where the binomial name is used.

was changed to …

Sequences are named using the common name for each species, except for P. brachycephalum.

7. PLOS authors have the option to publish the peer review history of their article. If published, this will include your full peer review and any attached files.

Do you want your identity to be public for this peer review? For information about this choice, including consent withdrawal, please see our Privacy Policy.

Reviewer #1: No

Reviewer #2: No

---

## [Editor Report · Decision Letter 2]

19 Nov 2020

Antifreeze protein dispersion in eelpouts and related fishes reveals migration and climate alteration within the last 20 Ma

PONE-D-19-34926R2

Dear Dr. Davies,

We’re pleased to inform you that your manuscript has been judged scientifically suitable for publication and will be formally accepted for publication once it meets all outstanding technical requirements.

Kind regards,

Michael Schubert

Academic Editor

PLOS ONE

---

## [Editor Report · Acceptance letter]

26 Nov 2020

PONE-D-19-34926R2 

Antifreeze protein dispersion in eelpouts and related fishes reveals migration and climate alteration within the last 20 Ma 

Dear Dr. Davies:

I'm pleased to inform you that your manuscript has been deemed suitable for publication in PLOS ONE. Congratulations! Your manuscript is now with our production department. 

Kind regards, 

on behalf of

Dr. Michael Schubert 

Academic Editor

PLOS ONE